# VoxDet: Rethinking 3D Semantic Scene Completion as Dense Object Detection

**Wuyang Li**[1]    **Zhu Yu**[2]    **Alexandre Alahi**[1]

[2]Zhejiang University       [1]École Polytechnique Fédérale de Lausanne (EPFL)

wuyang.li@epfl.ch    yu_zhu@zju.edu.cn    alexandre.alahi@epfl.ch

Project Page: https://vita-epfl.github.io/VoxDet/

## Abstract

Semantic Scene Completion (SSC) aims to reconstruct the 3D geometry and semantics of the surrounding environment. With dense voxel labels, prior works typically formulate SSC as a *dense segmentation task*, independently classifying each voxel. However, this paradigm neglects critical instance-centric discriminability, leading to instance-level incompleteness and adjacent ambiguities. To address this, we highlight a "free lunch" of SSC labels: the voxel-level class label has implicitly told the instance-level insight, which is ever-overlooked by the community. Motivated by this observation, we first introduce a training-free **Voxel-to-Instance (VoxNT) trick**: a simple yet effective method that freely converts voxel-level class labels into instance-level offset labels. Building on this, we further propose **VoxDet**, an instance-centric framework that reformulates the voxel-level SSC as *dense object detection* by decoupling it into two sub-tasks: offset regression and semantic prediction. Specifically, based on the lifted 3D volume, VoxDet first uses (a) Spatially-decoupled Voxel Encoder to generate disentangled feature volumes for the two sub-tasks, which learn task-specific spatial deformation in the densely projected tri-perceptive space. Then, we deploy (b) Task-decoupled Dense Predictor to address SSC via dense detection. Here, we first regress a 4D offset field to estimate distances (6 directions) between voxels and the corresponding object boundaries in the voxel space. The regressed offsets are then used to guide the instance-level aggregation in the classification branch, achieving instance-aware scene completion. VoxDet can be deployed on both camera and LiDAR input and jointly achieves state-of-the-art results on both benchmarks, which gives 63.0 IoU on the SemanticKITTI test set, **ranking 1$^{st}$** on the online leaderboard.

## 1   Introduction

Spatial AI [12] is crucial for autonomous systems to perceive and interpret the physical world. As a critical step, precise reconstruction of geometric structures and semantics lays the foundation for scene understanding [27, 41, 31], underpinning the downstream forecasting and planning [1, 14]. This capability is indispensable for applications such as autonomous driving and robotic navigation.

To this end, Semantic Scene Completion (SSC) has attracted significant attention by simultaneously inferring complete 3D geometry and semantics via occupancy representation. Prior SSC works can be broadly categorized into LiDAR-based [55, 50, 66] and camera-based approaches [83, 34, 79, 22]. The former uses sparse 3D inputs (e.g., point clouds) for SSC to provide precise geometric information. In contrast, camera-based methods have recently demonstrated promising potential due to their flexibility and computational efficiency. They employ dedicated vision-centric algorithms to lift 2D imagery into 3D space, enabling SSC, including Features Line-of-Sight Projection (FLoSP) [6],

39th Conference on Neural Information Processing Systems (NeurIPS 2025).

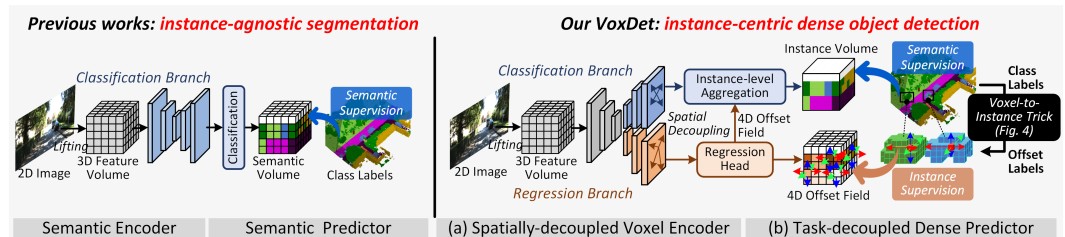

Figure 1: Schematic comparison of previous SSC paradigm [6, 2, 79] and the proposed VoxDet. **Left:** Existing works formulate SSC as 3D semantic segmentation, directly predicting voxel labels agnostic to object instances. **Right:** VoxDet reformulates SSC as dense object detection to explicitly learn with instances, which decouples it into instance-aware regression and classification. This is achieved via a Voxel-to-Instance (VoxNT) trick (Fig. 4), inspired by our observed free-lunch in labels (Sec. 3).

depth-driven back projection [34, 22, 78], and Lift-Splat-Shoot (LSS) [46] based voxels [79, 24, 2]. These methods offer unique advantages in resource-constrained settings such as robotics, owing to their low cost, high flexibility, and real-time processing capabilities.

While achieving great progress, existing methods uniformly formulate SSC as a semantic segmentation task[1] on dense voxels, which fail to understand the instance-level semantics and geometry. This paradigm typically employs 3D encoders [83, 79] to encode semantic patterns from the lifted volumes (see Fig. 1 **Left**), subsequently classifying each voxel independently. In SSC, due to the absent instance labels, this voxel-centric paradigm may seem viable, but it has to face severe instance-level incompleteness and adjacent ambiguities (see Fig. 7). Although prior works, such as Symphonize [22, 58, 40], have noticed this issue and attempted to mitigate it with object queries from 2D images, they still optimize the 3D space via dense segmentation on each voxel. The gap between the image and voxel domains prevents effective instance-driven learning, making the segmentation-based formulation fail to infer the complex environment with dynamic agents.

To address this issue, we first highlight a "free-lunch" of SSC labels: ***the voxel-level class label has implicitly told the instance-level insight, which is ever-overlooked by the community.*** First, unlike in 2D images where occlusion causes overlapping objects to conflate into a single entity, instances in 3D voxels are inherently separable. Every voxel is deterministically assigned to a single class without occlusion-induced ambiguity, making the instance-level discovery essentially realistic. Second, instance boundaries in 3D voxel space are naturally tractable and regressible, ensuring clear distinctions between instances. More details can be found in Sec. 3.

Inspired by these observations, we first develop a **Voxel-to-Instance (VoxNT) Trick** (Fig. 4) that can freely convert the voxel-level class labels to the instance-level offset labels, fully harnessing the free-lunch (Sec. 3). Here, the offset is defined as the Euclidean distance between each voxel and its associated instance borders. Then, these *free offset labels* prompt us to rethink the segmentation-based SSC formulation. In response to this, we propose **VoxDet** to reformulate Voxel-level SSC as instance-centric object Detection (see Fig. 1 **Right**). Unlike prior works, VoxDet decouples SSC into two sub-tasks: offset regression and semantic prediction. We achieve this with (a) Spatially-decoupled Voxel Encoder (SVE) and (b) Task-decoupled Dense Predictor (TDP) at the representation and prediction levels, respectively. Specifically, given 3D volumes lifted from 2D images [46], SVE first learns task-specific features by spatially deforming in the tri-perceptive space, which are sent to the two branches of TDP, respectively. Within TDP, we regress each voxel to the associated instance boundary by predicting a 4D offset field, which guides an instance-level aggregation for SSC.

*Hence, with our new detection-based formulation, VoxDet is the first work to achieve true instance-centric perception solely using voxel-level labels*, which is not only highly efficient but also achieves state-of-the-art performance on benchmarks, setting new records of 18.47/ 21.40 mIoU and 47.27/ 48.59 IoU on the SemanticKITTI and SSCBench-KITTI-360 datasets, respectively.

In summary, our contributions lie in the following aspects:

- By analyzing the difference of 2D pixels and 3D voxels (Sec. 3), we reveal the overlooked free lunch of SSC labels for instance-level learning. With this observation, we propose a

---

[1]For simplicity, in SSC, we uniformly refer to voxel recognition and completion [6] as dense classification.

VoxNT trick to freely convert voxel-level labels to instance-level offsets. As a byproduct, this trick can also identify wrong labels of dynamic objects (see Appx. B and Fig. 7).

- We reformulate SSC as a dense object detection task by advancing VoxDet, decoupling SSC into two sub-tasks: offset regression and semantic prediction, for instance-level SSC.

- We design a Spatially-decoupled Voxel Encoder (SVE) to decouple 3D volumes for our new SSC formulation, which learns task-specific spatial deformation in the densely projected tri-perceptive space for the two sub-tasks, avoiding the misalignment between them.

- We propose a Task-decoupled Dense Predictor (TDP) to enable instance-driven SSC. This comprises a regression branch that predicts a 4D offset field, delineating instance boundaries, and a classification branch using learned offsets for instance-aware aggregation.

## 2 Related Work

**3D Semantic Scene Completion** [55, 7] aims to jointly reconstruct the semantics and geometry of a surrounding environment with voxelization. Existing studies can be generally divided into LiDAR-based and camera-based methods [51]. The former utilizes point clouds to achieve high accuracy with precise depth completion [70–72], which is limited by the computational cost. Camera-based methods rely solely on 2D visual inputs to generate 3D scene understanding. With the advancement of monocular vision like LSS [46], these approaches offer great advantages in efficiency. MonoScene [6] pioneered the field of camera-based SSC by connecting 2D images with the 3D voxel space via the FLoSP. VoxFormer [34] uses 3D-to-2D back-projection and disseminates the semantics of the visible queries across the entire 3D volume via MAE [19]. Subsequent works have further enhanced voxel representations using techniques such as tri-perceptive enhancement [79, 2], diffusion models [35], vision-language models [58, 77], and extra modalities [18, 60]. However, existing methods uniformly treat SSC as a voxel segmentation, which lacks instance-level perception. In contrast, we reformulate SSC as a dense object detection task with explicit instance-level awareness.

**Dense Object Detection**, such as point-based FCOS [56, 62], is a fully convolutional paradigm known for its lightweight design, efficiency, and performance, which garnered significant attention prior to the DETR series [8, 63, 86, 43]. The core insight is the notion of "*densely detecting like segmentation*": every pixel within a bounding box regresses its distances in four directions (up, down, left, right) while simultaneously predicting the instance-level class label, which is a dense process like the per-pixel segmentation. The following works focus on improving this paradigm from different aspects, including considering better label assignment [81, 80] like ATSS [81], architecture search [59], spatially task decoupling [10], border enhancement [47], dense feature distillation [84, 73, 30, 29], optimization signals [68, 39]. Besides, dense detection has been extended to 3D vision. For instance, FCOS-3D [62] predicts 3D targets in the 2D space by regressing 3D attributes for each pixel, and UVTR [32] predicts a 3D position for each pixel to enhance instance-level localization. Additionally, some instance detection works propose a 3D-inspired paradigm for geometrically reliable detection in 2D views [26, 53]. Unlike previous works, we reverse the philosophy via a "*segmenting like dense detection*" framework that endows voxel-based segmentation with instance-level awareness, which *requires no additional manually annotated instance-level labels* for free instance-level learning.

## 3 Preliminaries and Motivation

We start by analyzing the differences between 2D pixel and 3D voxel space with respect to semantic-level classification and instance-level regression, from the perspective of dense perception [56]. We then clarify our observations and insights on the ever-overlooked "free-lunch" in voxel-level SSC labels and explain the associated motivation for the following methodology.

**Semantic-based Dense Classification.** In Fig. 2 (a,c), it can be observed that occlusion in the 2D space often leads to the merging of distinct object instances (e.g., the blue-highlighted cars) into a single entity (Fig. 2a). Due to the perceptive projection [13], multiple 3D points at varying depths in the world coordinate system converge into a single pixel of the image. In contrast, the 3D voxel space inherently avoids most of such occlusion, as each voxel is uniquely assigned a class label, including the *empty* class (Fig. 2c). *Consequently, the occlusion-free property of 3D voxels facilitates the natural separation of object instances*, although minor ambiguities may arise in cases such as the densely overlapping foliage of adjacent trees, which typically do not affect the scene understanding.

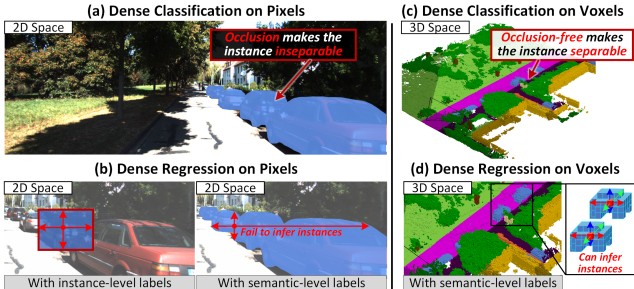

Figure 2: Conceptual comparison between 2D pixels and 3D voxels regarding semantic-level dense classification and instance-level regression. In **(a,b)**, 2D occlusion makes instance perception fail with semantic labels. In **(c,d)**, the occlusion-free nature of 3D space enables instances separable only using semantic labels.

**Instance-based Dense Regression.** In Fig. 2 (b,d), each pixel and voxel aims to regress the distance to the instance border [56]. In the 2D domain (Fig. 2b), such regression supports instance-level perception via bounding box labels; however, it fails to infer instance cues with semantic labels. *Conversely, in the 3D voxel space, the natural separability of instances makes it possible to regress instances without instance labels* (Fig. 2d). For example, consider the red voxel inside the highlighted car in the figure. As the voxel space is occlusion-free, we can regress from this voxel to its instance boundary by identifying adjacent voxels with differing semantic labels (see Fig. 4). This property enables us to infer instances using only semantic annotations, a valuable yet overlooked *free lunch*.

**Motivation and Insight**. Inspired by these observations, we aim to rethink Voxel-level SSC as object Detection [56], termed VoxDet, which adopts a novel "segmenting like dense detection" philosophy. Instead of directly recognizing each independent voxel [83, 79, 2, 34], VoxDet decouples SSC into regression and classification sub-tasks in the 3D voxel space, where, in particular, the core innovative regression (or offset) branch enables explicit instance-level 3D regression. The learned offsets are subsequently used for instance-level aggregation, facilitating scene understanding. Note that this fundamentally differs from and advances beyond the traditional segmentation paradigm by *explicitly lifting voxels to instances*, with potential applicability to the broader occupancy community.

## 4 Methodology

**Overview.** Fig. 3 shows the overall workflow of our VoxDet. Given RGB input, we follow previous works [79] to conduct **(a)** 2D-to-3D lifting to generate 3D feature volumes $\mathbf{V}$, which uses the estimated depth $\mathbf{Z}$ given by the arbitrary depth estimator [5, 52]. Then, we send $\mathbf{V}$ to **(b)** Spatially-decoupled Voxel Encoder to generate the disentangled feature for dense classification $\mathbf{V}_{\text{cls}}$ and regression $\mathbf{V}_{\text{reg}}$. Here, we encourage the two tasks to learn spatially decoupled features to avoid task-misalignment [68], which is achieved in a densely projected tri-perceptive (TPV) space. Next, the decoupled volumes are sent to **(c)** Task-decoupled Dense Predictor. In this part, we first regress a 4D offset field $\Delta$ to estimate the distance between each voxel $\mathbf{V}_{i,j,k}$ to the instance boundary $\Delta_{i,j,k} \in \mathbb{R}^6$ in six directions (see Fig. 4). The learned offset $\Delta$ is subsequently sent to the classification branch to guide instance-level aggregation, thereby achieving instance-aware semantic scene completion.

### 4.1 2D-to-3D Lifting

We follow previous SSC works [79, 2, 34] to conduct the same 2D-to-3D lifting (Fig. 3a), outputting the 3D feature volume $\mathbf{V} \in \mathbb{R}^{X \times Y \times Z \times C}$, where $X$, $Y$, $Z$, and $C$ is the depth, width, height, and channel respectively. The process is briefly described as follows. More details are in Appx. C.

Given the RGB image $\mathbf{I}$, we extract the 2D image feature $\mathbf{F}^{2D}$ using the image encoder, and estimate the depth map $\mathbf{Z}$ with the frozen depth estimator [5, 52] following the unified SSC practice [79, 34, 2]. Then, we estimate the depth probability $\mathbf{D}$ using LSS [46]. Based on these, we can establish the 3D feature $\mathbf{F}^{3D}$ using the fused depth probability $\mathbf{D}$ and pre-extracted 2D feature $\mathbf{F}^{2D}$ [79], which is subsequently projected onto the voxel grid to generate the 3D volume $\mathbf{V}$. In this procedure, each voxel in $\mathbf{V}$ is able to query the information from 3D features $\mathbf{F}^{3D}$ via deformable cross-attention [11].

### 4.2 Spatially-decoupled Voxel Encoder

Given the 3D volume $\mathbf{V}$, we aim to extract task-specific representations for the two tasks. Prior encoders are designed for voxel segmentation, which lacks the spatial context for regression with

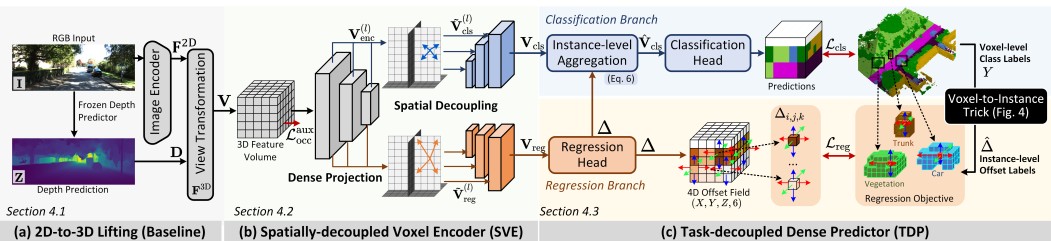

Figure 3: Overview of our VoxDet. After 2D-to-3D lifting, VoxDet spatially decouples 3D volumes $\mathbf{V}$ into two task-specific branches, learning different spatial deformations in the densely projected tri-perceptive space. Then, VoxDet regresses a 4D offset field $\Delta$ towards instance boundaries with $\mathbf{V}_{\text{reg}}$, serving for the instance-level aggregation with $\mathbf{V}_{\text{cls}}$ in the classification branch.

task-misalignment [54, 28, 10]. Hence, we spatially decouple $\mathbf{V}$ into $\mathbf{V}_{\text{cls}}$ for classification and $\mathbf{V}_{\text{reg}}$ for regression (Fig. 3b) in a densely projected TPV space. Fig. 6 shows the decoupling effect.

**Dense Projection.** Given the 3D volume $\mathbf{V}$, we use a shared encoder to extract $L$-level task-shared feature volumes $\{\mathbf{V}_{\text{enc}}^{(l)}\}_{l=1}^L$. Due to the task-misaligned feature preference [68], directly using the shared volume for the two tasks will severely influence the instance-level perception. To resolve this, we spatially decouple task-specific features in a densely projected tri-perceptive (TPV) space [21]. This can avoid the large computational cost of decoupling in 3D, effectively reducing the dimension to 2D. To this end, a dense projection $\mathcal{P}_d(\cdot)$ is proposed to generate 2D feature planes for our detection-based formulation. Specifically, given $\mathbf{V}_{\text{enc}}^{(l)}$, we learn per-voxel dense weight with a linear layer $\boldsymbol{\eta} = \mathbf{W}_d(\mathbf{V}_{\text{enc}}^{(l)}), \boldsymbol{\eta} \in \mathbb{R}^{X \times Y \times Z \times 3}$, which guides the adaptive pooling for the three TPV planes. Then, we conduct pooling across each axis $(X, Y, Z)$ to generate projected 2D feature planes:

$$\mathcal{P}_d^Z, \mathcal{P}_d^Y, \mathcal{P}_d^X(\mathbf{V}) = \sum_{z=1}^Z \mathcal{S}_Z(\boldsymbol{\eta}_{[:,:,:,1]}) \circ \mathbf{V}_{[:,:,z]}, \sum_{y=1}^Y \mathcal{S}_Y(\boldsymbol{\eta}_{[:,:,:,2]}) \circ \mathbf{V}_{[:,y,:]}, \sum_{x=1}^X \mathcal{S}_X(\boldsymbol{\eta}_{[:,:,:,3]}) \circ \mathbf{V}_{[x,:,:]}.$$
(1)

Here, $\mathcal{P}_d^{(A)}$ indicates the $(A)$-axis projection, $\mathcal{S}_{(A)}$ is softmax on the $(A)$ axis, and $\circ$ is Hadamard product. Next, we deploy a 2D refinement layer to generate TPV feature at $XY, XZ, YZ$ planes:

$$\begin{aligned}
\mathbf{V}_{\text{cls}}^{XY} &= \text{Conv}_{\text{cls}}^{XY}(\mathcal{P}_d^Z(\mathbf{V}_{\text{enc}}^{(l)})), \mathbf{V}_{\text{cls}}^{XZ} = \text{Conv}_{\text{cls}}^{XZ}(\mathcal{P}_d^Y(\mathbf{V}_{\text{enc}}^{(l)})), \mathbf{V}_{\text{cls}}^{YZ} = \text{Conv}_{\text{cls}}^{YZ}(\mathcal{P}_d^X(\mathbf{V}_{\text{enc}}^{(l)})), \\
\mathbf{V}_{\text{reg}}^{XY} &= \text{Conv}_{\text{reg}}^{XY}(\mathcal{P}_d^Z(\mathbf{V}_{\text{enc}}^{(l)})), \mathbf{V}_{\text{reg}}^{XZ} = \text{Conv}_{\text{reg}}^{XZ}(\mathcal{P}_d^Y(\mathbf{V}_{\text{enc}}^{(l)})), \mathbf{V}_{\text{reg}}^{YZ} = \text{Conv}_{\text{reg}}^{YZ}(\mathcal{P}_{\text{ada}}^X(\mathbf{V}_{\text{enc}}^{(l)})).
\end{aligned}$$
(2)

Here, $\text{Conv}_{(T)}^{(P)}(\cdot)$ is 2D convolution module for plane $(P)$ and task $(T)$. Compared with the conventional TPV pooling [21], our dense projection can adaptively discover the essential voxels during the 3D-to-2D dimensional reduction, thereby mitigating the information loss for the dense perception.

**Spatial Decoupling.** With projected 2D features, we spatially decouple it in each TPV plane with different spatial offsets, encouraging two tasks to focus on task-specific regions. Then, we use a Conv layer to fuse the decoupled features with the same expanded size, which is carried out as follows:

$$\begin{aligned}
\tilde{\mathbf{V}}_{\text{cls}}^{(l)} &= \text{Conv}_{\text{cls}}^{\text{fuse}}\left(\text{DefConv}_{\text{cls}}^{XY}(\mathbf{V}_{\text{cls}}^{XY}) + \text{DefConv}_{\text{cls}}^{XZ}(\mathbf{V}_{\text{cls}}^{XZ}) + \text{DefConv}_{\text{cls}}^{YZ}(\mathbf{V}_{\text{cls}}^{YZ})\right); \\
\tilde{\mathbf{V}}_{\text{reg}}^{(l)} &= \text{Conv}_{\text{reg}}^{\text{fuse}}\left(\text{DefConv}_{\text{reg}}^{XY}(\mathbf{V}_{\text{reg}}^{XY}) + \text{DefConv}_{\text{reg}}^{XZ}(\mathbf{V}_{\text{reg}}^{XZ}) + \text{DefConv}_{\text{reg}}^{YZ}(\mathbf{V}_{\text{reg}}^{YZ})\right).
\end{aligned}$$
(3)

Here, $\text{DefConv}(\cdot)$ is the 2D deformable convolution module, which spatially decouples the dense voxel features for the two tasks. This decoupling reduces the computational burden and preserves task-driven spatial context, effectively addressing the misaligned feature preference of the two tasks. Finally, we send the decoupled multi-level features to the lightweight FPN [38] branches, respectively, to improve the task-specific learning, and collect the last-layer output with the same resolution as $\mathbf{V}$ for the two tasks, which is denoted as: $\mathbf{V}_{\text{cls}} = \text{FPN}(\{\tilde{\mathbf{V}}_{\text{cls}}^{(l)}\}_{l=1}^L)$ and $\mathbf{V}_{\text{reg}} = \text{FPN}(\{\tilde{\mathbf{V}}_{\text{reg}}^{(l)}\}_{l=1}^L)$.

### 4.3 Task-decoupled Dense Predictor

Then, the decoupled features $\mathbf{V}_{\text{cls}}$ and $\mathbf{V}_{\text{reg}}$ are sent to the classification and regression branches (see Fig. 3c). Here, we first densely regress the instance boundary to identify 3D objects, then aggregate instance-level semantics in the classification branch, achieving an instance-centric SSC prediction.

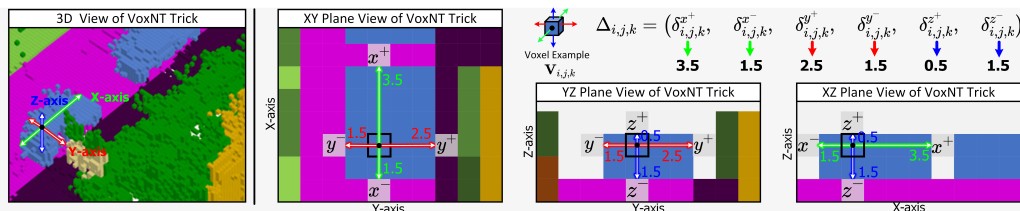

Figure 4: Illustration of our regression objective. For each voxel $\mathbf{V}_{i,j,k}$, we scan in 6 directions and calculate the distance to the instance boundary to generate labels $\hat{\Delta}_{i,j,k}$, using the free-lunch (Sec. 3).

**Regression Branch.** Given the volume for regression $\mathbf{V}_{\text{reg}} \in \mathbb{R}^{X \times Y \times Z \times C}$, we aim to regress the distance to the instance boundary for each voxel $\mathbf{V}_{i,j,k} \in \mathbf{V}_{\text{reg}}$ (see Fig. 4) in six directions, which form a 4D offset field $\Delta \in \mathbb{R}^{X \times Y \times Z \times 6}$. We use 6 directions as it is the minimum number to determine a 3D bounding box, similar to the 2D bounding boxes determined with 4 directions [56]. Thus, considering the voxel in $\mathbf{V}_{\text{reg}}$ at position $(i, j, k)$, the spatially associated element $\Delta_{i,j,k}$ in the offset field represents a 6-channel vector for the specific offset $\delta$ along the $X$, $Y$, and $Z$ axes, which is divided into positive ($^+$) and negative ($^-$) directions. This can be denoted as follows,

$$\Delta = \text{RegressionHead}(\mathbf{V}_{\text{reg}}), \text{ where } \Delta_{i,j,k} = \left( \delta_{i,j,k}^{x^+}, \delta_{i,j,k}^{x^-}, \delta_{i,j,k}^{y^+}, \delta_{i,j,k}^{y^-}, \delta_{i,j,k}^{z^+}, \delta_{i,j,k}^{z^-} \right) \in \mathbb{R}^6. \quad (4)$$

Here, $\text{RegressionHead}(\cdot)$ is a lightweight head to predict the offset field, which is simply deployed as a two-convolution module to enable non-linearity with 6-channel output, followed by `Sigmoid` for normalization. Compared with the anchor-based design [48], this design is highly computationally efficient due to its fully convolutional nature when deployed in the dense 3D voxel space.

**Regression Loss.** In SSC, we only have dense class labels for the voxel grid $Y \in \mathbb{N}^{X \times Y \times Z}$ without instance-level labels for regression, such as bounding boxes. Hence, to break through this, we propose a **Voxel-to-Instance (VoxNT) Trick** to freely transform the class labels to the instance-level offsets, fully using the free lunch in Sec. 3. The algorithm details are in the Appendix.

Fig. 4 shows the VoxNT trick. In brief, for each voxel $\Delta_{i,j,k}$, e.g., the blue example, we scan across 6 directions $\mathbf{d} \in \{x^+, x^-, y^+, y^-, z^+, z^-\}$ in labels $Y$, and stop when the class of next scanned voxel changes, indicating approaching the border. Then, we save the scanning distance as the offset labels: $\hat{\Delta} \in \mathbb{R}^{X \times Y \times Z \times 6}$. For stability, we round up $\hat{\Delta}$ to the integer and normalize into $[0, 1]$ via the volume size. Finally, we deploy L1 loss to optimize the regression with enough gradient:

$$\mathcal{L}_{\text{reg}} = \Sigma_{i,j,k,d}^{X,Y,Z,6} |\Delta_{i,j,k,d} - \hat{\Delta}_{i,j,k,d}|, \quad (5)$$

where $\Delta \in \mathbb{R}^{X \times Y \times Z \times 6}$ is the predicted 4D offset field. Surprisingly, as a by-product, the offset field can identify wrongly labeled dynamic objects (like cars) for more accurate training (see Appx. B.3).

**Classification Branch.** Based on the regressed offset $\Delta$, we aim to enhance instance-level perception by aggregating the semantics within instances. A natural idea [17] is to crop 3D bounding boxes, which is extremely computationally expensive. Hence, we propose an alternative solution by adaptively aggregating the voxels at regressed positions, considering the informative nature of borders [23, 85]. For each voxel $\mathbf{V}_{i,j,k} \in \mathbf{V}_{\text{cls}}$, we have the predicted offsets $\Delta_{i,j,k}$ associating with 6 voxels, which are extracted as a instance-level voxel set $\mathbf{V}_{\delta}$, where $\delta \in \Delta_{i,j,k}$ and $|\Delta_{i,j,k}|=6$. Then, each voxel $\mathbf{V}_{i,j,k}$ will query the semantics from the associated voxel set $\mathbf{V}_{\delta}$ attentively:

$$\hat{\mathbf{V}}_{i,j,k} = \text{Norm}\Big( \sum_{\delta \in \Delta_{i,j,k}} \frac{\exp\big(\mathbf{W}_q \mathbf{V}_{i,j,k}\big)^\top \big(\mathbf{W}_k \mathbf{V}_\delta / \sqrt{d_k}\big)}{\sum_{\delta' \in \Delta_{i,j,k}} \exp\big(\mathbf{W}_q \mathbf{V}_{i,j,k}\big)^\top \big(\mathbf{W}_k \mathbf{V}_{\delta'} / \sqrt{d_k}\big)} \mathbf{W}_v \mathbf{V}_\delta + \mathbf{V}_{i,j,k}\Big). \quad (6)$$

Here, $\hat{\mathbf{V}}_{i,j,k} \in \hat{\mathbf{V}}_{\text{cls}}$ is refined feature, each $\mathbf{W}_{(\cdot)}$ is a linear layer, $d_k$ is the channel number, and Norm is `Group Normalization`. With $N = 4$ aggregation layers, each voxel is able to gain sufficient perception of the whole instance, enhancing instance-level semantics. This design is justified in Fig. 5. Finally, $\hat{\mathbf{V}}_{\text{cls}}$ is sent to a simple classification head to generate class predictions.

**Classification Loss.** In dense detection [56], classification is optimized as segmentation in a per-pixel dense manner. Hence, to optimize the classification branch in our VoxDet, we naturally deploy the dense semantic prediction loss following the previous SSC arts [83, 79, 82]:

$$\mathcal{L}_{\text{cls}} = \mathcal{L}_{\text{ce}} + \mathcal{L}_{\text{aff}}^{\text{bin}} + \mathcal{L}_{\text{aff}}^{\text{sem}}, \quad (7)$$

where $\mathcal{L}_{\text{ce}}$ is the cross-entropy loss weighted by class frequencies, and $\mathcal{L}_{\text{aff}}^{\text{bin}}$ and $\mathcal{L}_{\text{aff}}^{\text{sem}}$ are affinity loss with binary and semantic settings. We set the consistent loss weight as 1.0 for convenience.

Table 1: Results of SSC on SemanticKITTI [3] hidden test set. * is from [21, 82]. **T** is using extra temporal information. The best and the second best results are in **bold** and underlined, respectively.

| Method | T | IoU | mIoU | road | sidewalk | parking | other-grnd. | building | car | truck | bicycle | motorcycle | other-veh. | vegetation | trunk | terrain | person | bicyclist | motorcyclist | fence | pole | traf.-sign |
|---|---|---|---|---|---|---|---|---|---|---|---|---|---|---|---|---|---|---|---|---|---|---|
| MonoScene* [6] | | 34.16 | 11.08 | 54.70 | 27.10 | 24.80 | 5.70 | 14.40 | 18.80 | 3.30 | 0.50 | 0.70 | 4.40 | 14.90 | 2.40 | 19.50 | 1.00 | 1.40 | 0.40 | 11.10 | 3.30 | 2.10 |
| TPVFormer [21] | | 34.25 | 11.26 | 55.10 | 27.20 | 27.40 | 6.50 | 14.80 | 19.20 | 3.70 | 1.00 | 0.50 | 2.30 | 13.90 | 2.60 | 20.40 | 1.10 | 2.40 | 0.30 | 11.00 | 2.90 | 1.50 |
| SurroundOcc [65] | | 34.72 | 11.86 | 56.90 | 28.30 | 30.20 | 6.80 | 15.20 | 20.60 | 1.40 | 1.60 | 1.20 | 4.40 | 14.90 | 3.40 | 19.30 | 1.40 | 2.00 | 0.10 | 11.30 | 3.90 | 2.40 |
| OccFormer [82] | | 34.53 | 12.32 | 55.90 | 30.30 | 31.50 | 6.50 | 15.70 | 21.60 | 1.20 | 1.50 | 1.70 | 3.20 | 16.80 | 3.90 | 21.30 | 2.20 | 1.10 | 0.20 | 11.90 | 3.80 | 3.70 |
| IAMSSC [67] | | 43.74 | 12.37 | 54.00 | 25.50 | 24.70 | 6.90 | 19.20 | 21.30 | 3.80 | 1.10 | 0.60 | 3.90 | 22.70 | 5.80 | 19.40 | 1.50 | 2.90 | 0.50 | 11.90 | 5.30 | 4.10 |
| VoxFormer [34] | | 42.95 | 12.20 | 53.90 | 25.30 | 21.10 | 5.60 | 19.80 | 20.80 | 3.50 | 1.00 | 0.70 | 3.70 | 22.40 | 7.50 | 21.30 | 1.40 | 2.60 | 0.20 | 11.10 | 5.10 | 4.90 |
| VoxFormer [34] | ✓ | 43.21 | 13.41 | 54.10 | 26.90 | 25.10 | 7.30 | 23.50 | 21.70 | 3.60 | 1.90 | 1.60 | 4.10 | 24.40 | 8.10 | 24.20 | 1.60 | 1.10 | 0.00 | 13.10 | 6.60 | 5.70 |
| DepthSSC [76] | | 44.58 | 13.11 | 55.64 | 27.25 | 25.72 | 5.78 | 20.46 | 21.94 | 3.74 | 1.35 | 0.98 | 4.17 | 23.37 | 7.64 | 21.56 | 1.34 | 2.79 | 0.28 | 12.94 | 5.87 | 6.23 |
| Symphonize [22] | | 42.19 | 15.04 | 58.40 | 29.30 | 26.90 | 11.70 | 24.70 | 23.60 | 3.20 | 3.60 | 2.60 | 5.60 | 24.20 | 10.00 | 23.10 | **3.20** | 1.90 | **2.00** | 16.10 | 7.70 | 8.00 |
| HASSC [61] | | 43.40 | 13.34 | 54.60 | 27.70 | 23.80 | 6.20 | 21.10 | 22.80 | 4.70 | 1.60 | 1.00 | 3.90 | 23.80 | 8.50 | 23.30 | 1.60 | **4.00** | 0.30 | 13.10 | 5.80 | 5.50 |
| HASSC [61] | ✓ | 42.87 | 14.38 | 55.30 | 29.60 | 25.90 | 11.30 | 23.10 | 23.00 | 2.90 | 1.90 | 1.50 | 4.90 | 24.80 | 9.80 | 26.50 | 1.40 | 3.00 | 0.00 | 14.30 | 7.00 | 7.10 |
| StereoScene [25] | | 43.34 | 15.36 | 61.90 | 31.20 | 30.70 | 10.70 | 24.20 | 22.80 | 2.80 | 3.40 | 2.40 | **6.10** | 23.80 | 8.40 | 27.00 | 2.90 | 2.20 | 0.50 | 16.50 | 7.00 | 7.20 |
| H2GFormer [64] | | 44.20 | 13.72 | 56.40 | 28.60 | 26.50 | 4.90 | 22.80 | 23.40 | 4.80 | 0.80 | 0.90 | 4.10 | 24.60 | 9.10 | 23.80 | 1.20 | 2.50 | 0.10 | 13.30 | 6.40 | 6.30 |
| H2GFormer [64] | ✓ | 43.52 | 14.60 | 57.90 | 30.40 | 30.00 | 6.90 | 24.00 | 23.70 | 5.20 | 0.60 | 1.20 | 5.00 | 25.20 | 10.70 | 25.80 | 1.10 | 0.10 | 0.00 | 14.60 | 7.50 | 9.30 |
| MonoOcc [83] | | - | 13.80 | 55.20 | 27.80 | 25.10 | 9.70 | 21.40 | 23.20 | 5.20 | 2.20 | 1.50 | 5.40 | 24.00 | 8.70 | 23.00 | 1.70 | 2.00 | 0.20 | 13.40 | 5.80 | 6.40 |
| CGFormer [79] | | 44.41 | 16.63 | 64.30 | 34.20 | 34.10 | 12.10 | 25.80 | 26.10 | 4.30 | 3.70 | 1.30 | 2.70 | 24.50 | 11.20 | 29.30 | 1.70 | 3.60 | 0.40 | 18.70 | 8.70 | 9.30 |
| HTCL [24] | ✓ | 44.23 | 17.09 | 64.40 | 34.80 | 33.80 | 12.40 | 25.90 | **27.30** | 5.70 | 1.80 | 2.20 | 5.40 | 25.30 | 10.80 | 31.20 | 1.10 | 3.10 | 0.90 | 21.10 | 9.00 | 8.30 |
| **VoxDet (Ours)** | | **47.27** | **18.47** | **64.70** | **35.50** | **34.80** | **14.40** | **28.10** | 26.90 | **6.10** | **5.90** | **5.10** | 5.00 | **28.70** | **13.60** | **31.70** | 3.10 | **4.00** | 1.30 | **21.50** | **10.10** | **10.30** |

Table 2: Results of SSC on SSCBench-KITTI360 test set.* is the corrected results using the consistent dataset version [18]. The best and second best results are are in **bold** and underlined, respectively.

| Method | IoU | mIoU | car | bicycle | motorcycle | truck | other-veh. | person | road | parking | sidewalk | other-grnd. | building | fence | vegetation | terrain | pole | traf.-sign | other-struct. | other-obj. |
|---|---|---|---|---|---|---|---|---|---|---|---|---|---|---|---|---|---|---|---|---|
| *LiDAR-based methods* | | | | | | | | | | | | | | | | | | | | |
| SSCNet [55] | 53.58 | 16.95 | 31.95 | 0.00 | 0.17 | 10.29 | 0.00 | 0.07 | 65.70 | 17.33 | 41.24 | 3.22 | 44.41 | 6.77 | 43.72 | 28.87 | 0.78 | 0.75 | 8.69 | 0.67 |
| LMSCNet [50] | 47.35 | 13.65 | 20.91 | 0.00 | 0.00 | 0.26 | 0.58 | 0.00 | 62.95 | 13.51 | 33.51 | 0.20 | 43.67 | 0.33 | 40.01 | 26.80 | 0.00 | 0.00 | 3.63 | 0.00 |
| *Camera-based methods* | | | | | | | | | | | | | | | | | | | | |
| MonoScene [6] | 37.87 | 12.31 | 19.34 | 0.43 | 0.58 | 8.02 | 2.03 | 0.86 | 48.35 | 11.38 | 28.13 | 3.32 | 32.89 | 3.53 | 26.15 | 16.75 | 6.92 | 5.67 | 4.20 | 3.09 |
| TPVFormer [21] | 40.22 | 13.64 | 21.56 | 1.09 | 1.37 | 8.06 | 2.57 | 2.38 | 52.99 | 11.99 | 31.07 | 3.78 | 34.83 | 4.80 | 30.08 | 17.52 | 7.46 | 5.86 | 5.48 | 2.70 |
| OccFormer [82] | 40.27 | 13.81 | 22.58 | 0.66 | 0.26 | 9.89 | 3.82 | 2.77 | 54.30 | 13.44 | 31.53 | 3.55 | 36.42 | 4.80 | 31.00 | 19.51 | 7.77 | 8.51 | 6.95 | 4.60 |
| VoxFormer [34] | 38.76 | 11.91 | 17.84 | 1.16 | 0.89 | 4.56 | 2.06 | 1.63 | 47.01 | 9.67 | 27.21 | 2.89 | 31.18 | 4.97 | 28.99 | 14.69 | 6.51 | 6.92 | 3.79 | 2.43 |
| IAMSSC [67] | 41.80 | 12.97 | 18.53 | 2.45 | 1.76 | 5.12 | 3.92 | 3.09 | 47.55 | 10.56 | 28.35 | 4.12 | 31.53 | 6.28 | 29.17 | 15.24 | 8.29 | 7.01 | 6.35 | 4.19 |
| DepthSSC [76] | 40.85 | 14.28 | 21.90 | 2.36 | 4.30 | 11.51 | 4.56 | 2.92 | 50.88 | 12.89 | 30.27 | 2.49 | 37.33 | 5.22 | 29.61 | 21.59 | 5.97 | 7.71 | 5.24 | 3.51 |
| Symphonies* [22] | 43.41 | 17.82 | 26.86 | 4.21 | 4.90 | 14.20 | 7.76 | 6.57 | 57.30 | 13.58 | 35.24 | 4.57 | 39.20 | 7.95 | 34.33 | 19.19 | 14.04 | 15.78 | 8.23 | 6.04 |
| SGN-S [44] | 46.22 | 17.71 | 28.20 | 2.09 | 3.02 | 11.95 | 3.68 | 4.20 | 59.49 | 14.50 | 36.53 | 4.24 | 39.79 | 7.14 | 36.61 | 23.10 | 14.86 | 16.14 | 8.24 | 4.95 |
| SGN-T [44] | 47.06 | 18.25 | 29.03 | 3.43 | 2.90 | 10.89 | 5.20 | 2.99 | 58.14 | 15.04 | 36.40 | 4.43 | 42.02 | 7.72 | 38.17 | 23.22 | 16.73 | 16.38 | 9.93 | 5.86 |
| CGFormer [79] | 48.07 | **20.05** | 29.85 | 3.42 | 3.96 | 17.59 | 6.79 | 6.63 | **63.85** | 17.15 | **40.72** | 5.53 | 42.73 | 8.22 | 38.80 | 24.94 | 16.24 | 17.45 | 10.18 | 6.77 |
| SGFormer [18] | 46.35 | 18.30 | 27.80 | 0.91 | 2.55 | 10.73 | 5.67 | 4.28 | 61.04 | 13.21 | 37.00 | 5.07 | 43.05 | 7.46 | 38.98 | 24.87 | 15.75 | 16.90 | 8.85 | 5.33 |
| **VoxDet (Ours)** | **48.59** | **21.40** | **29.92** | **5.13** | **8.36** | **19.13** | **8.04** | **7.84** | 62.83 | **18.99** | 40.10 | **5.58** | **44.47** | **10.62** | **39.03** | **26.16** | **18.19** | **20.78** | **11.66** | **8.34** |

## 4.4 Optimization

To train the proposed VoxDet, we implement the whole optimization objective written as follows,

$$\mathcal{L}_{\text{VoxDet}} = \mathcal{L}_{\text{cls}} + \mathcal{L}_{\text{reg}} + \lambda \mathcal{L}_{\text{occ}}^{\text{aux}}, \tag{8}$$

where $\mathcal{L}_{\text{cls}}$ is dense classification loss (Eq. 7) and $\mathcal{L}_{\text{reg}}$ is dense regression loss (Eq. 5). Following the baseline [83], we retain the voxel-centric segmentation loss (cross-entropy and affinity terms) applied to $\mathbf{V}$ as an auxiliary prior constraint ($\mathcal{L}_{\text{occ}}^{\text{aux}}$) before instance-centric learning, which can stabilize the optimization. This also enhances the multiple-run robustness, which is justified in the Appendix. The weighting factor $\lambda = 0.2$ is empirically adjusted to balance the model learning at the instance level.

## 5 Experiments

### 5.1 Comparison with State-of-the-art Methods

**Camaera-based SSC.** Tab. 1 shows the comparison on the hidden test set of SemanticKITTI [3]. VoxDet achieves new records with an IoU of 47.27 and mIoU of 18.47. Compared with the methods using additional temporal labels like VoxFormer-T [34], HASSC-T [61], H2GFormer-T [64], and HTCL-T [24], VoxDet comprehensively surpasses them and gives noticeable 3.04 and 1.38 gains on IoU and mIoU over the previous best entry [24]. We further list the comparison on SSCBench-KITTI-360 [36] in Tab. 2. VoxDet achieves an IoU of 48.59 and mIoU of 21.40, outperforming all

Table 3: Results of LiDAR-based SSC on SemanticKITTI [3] hidden test set with single frame (no extra temporal information) for fair comparison. Our VoxDet only uses point cloud input. The best and the second best results are in **bold** and underlined, respectively. Previous SoTA is [57].

| Method | IoU | mIoU | road | sidewalk | parking | other-grnd. | building | car | truck | bicycle | motorcycle | other-veh. | vegetation | trunk | terrain | person | bicyclist | motorcyclist | fence | pole | traf-sign |
|---|---|---|---|---|---|---|---|---|---|---|---|---|---|---|---|---|---|---|---|---|---|
| SSCNet [55] | 29.8 | 9.5 | 27.6 | 17.0 | 15.6 | 6.0 | 20.9 | 10.4 | 1.8 | 0.0 | 0.0 | 0.1 | 25.8 | 11.9 | 18.2 | 0.0 | 0.0 | 0.0 | 14.4 | 7.9 | 3.7 |
| SSCNet-full [55] | 50.0 | 16.1 | 51.2 | 30.8 | 27.1 | 6.4 | 34.5 | 24.3 | 1.2 | 0.5 | 0.8 | 4.3 | 35.3 | 18.2 | 29.0 | 0.3 | 0.3 | 0.0 | 19.9 | 13.1 | 6.7 |
| TS3D [15] | 29.8 | 9.5 | 28.0 | 17.0 | 15.7 | 4.9 | 23.2 | 10.7 | 2.4 | 0.0 | 0.0 | 0.2 | 24.7 | 12.5 | 18.3 | 0.0 | 0.1 | 0.0 | 13.2 | 7.0 | 3.5 |
| TS3D/DNet [4] | 25.0 | 10.2 | 27.5 | 18.5 | 18.9 | 6.6 | 22.1 | 8.0 | 2.2 | 0.1 | 0.0 | 4.0 | 19.5 | 12.9 | 20.2 | 2.3 | 0.6 | 0.0 | 15.8 | 7.6 | 7.0 |
| LMSCNet [50] | 55.3 | 17.0 | 64.0 | 33.1 | 24.9 | 3.2 | 38.7 | 29.5 | 2.5 | 0.0 | 0.0 | 0.1 | 40.5 | 19.0 | 30.8 | 0.0 | 0.0 | 0.0 | 20.5 | 15.7 | 0.5 |
| LMSCNet-SS [50] | 56.7 | 17.6 | 64.8 | 34.7 | 29.0 | 4.6 | 38.1 | 30.9 | 1.5 | 0.0 | 0.0 | 0.8 | 41.3 | 19.9 | 32.1 | 0.0 | 0.0 | 0.0 | 20.5 | 15.7 | 0.8 |
| Local-DIFs [49] | 57.7 | 22.7 | 67.9 | 42.9 | **40.1** | 11.4 | 40.4 | 34.8 | 4.4 | 3.6 | 2.4 | 4.8 | 42.2 | 26.5 | 39.1 | 2.5 | 1.1 | 0.0 | 29.0 | 21.3 | 17.5 |
| JS3C-Net [69] | 56.6 | 23.8 | 64.7 | 39.9 | 34.9 | 14.1 | 39.4 | 33.3 | **7.2** | **14.4** | **8.8** | **12.7** | 43.1 | 19.6 | 40.5 | **8.0** | **5.1** | 0.4 | 30.4 | 18.9 | 15.9 |
| SSA-SC [74] | 58.8 | 23.5 | 72.2 | 43.7 | 37.4 | 10.9 | 43.6 | 36.5 | 5.7 | 13.9 | 4.6 | 7.4 | 43.5 | 25.6 | 41.8 | 4.4 | 2.6 | 0.7 | 30.7 | 14.5 | 6.9 |
| L2COcc-D [60] | 45.3 | 18.1 | 68.2 | 36.9 | 34.6 | **16.2** | 25.8 | 28.3 | 4.5 | 4.9 | 3.3 | 7.2 | 26.2 | 11.9 | 32.0 | 2.1 | 2.4 | 0.3 | 21.6 | 9.6 | 9.5 |
| L2COcc-L [60] | 60.3 | 23.3 | 68.5 | 40.6 | 33.2 | 6.1 | 41.5 | 36.8 | 5.4 | 8.7 | 4.1 | 9.0 | 42.6 | 28.7 | 36.9 | 1.4 | 2.9 | 1.0 | 27.7 | 27.0 | 21.9 |
| VPNet [57] | 60.4 | 25.0 | 72.4 | **44.3** | 40.5 | 14.8 | 44.0 | 37.2 | 4.3 | 14.0 | **9.8** | 8.2 | 45.3 | **30.9** | 42.1 | 4.9 | 2.0 | **2.4** | 32.7 | 17.1 | 8.8 |
| **VoxDet-L (Ours)** | **63.0** | **26.0** | **73.0** | 43.6 | 37.5 | 10.3 | **44.5** | **37.7** | 6.6 | 9.9 | 6.2 | 11.8 | **45.9** | 30.7 | **43.5** | 2.7 | 3.2 | 1.3 | **34.0** | 27.8 | 23.7 |

Table 4: Ablation study on each key module.

| | TDP | | SVE | | IoU | mIoU | N_param |
|---|---|---|---|---|---|---|---|
| | REG | CLS | REG | CLS | (%) | (%) | (M) |
| (a) | | | | | 42.71 | 16.28 | 48.7 |
| (b) | ✓ | | | | 45.42 | 16.42 | 48.9 |
| (c) | ✓ | ✓ | | | 46.79 | 17.85 | 49.4 |
| (d) | ✓ | ✓ | ✓ | | 47.14 | 18.02 | 51.1 |
| (e) | ✓ | ✓ | | ✓ | 47.08 | 18.27 | 51.1 |
| (f) | ✓ | ✓ | ✓ | ✓ | **47.36** | **18.73** | 52.8 |

Table 5: Further analysis of varied designs.

| | Detailed designs | IoU (%) | mIoU (%) |
|---|---|---|---|
| TDP | $\Delta$ guidance $\rightarrow$ Self-attention | 46.82 | 18.15 |
| | Eq. 6 $\rightarrow$ Weighted fusion | 47.01 | 18.38 |
| SVE | Eq. 1 $\rightarrow$ Average pooling | 47.18 | 18.49 |
| | DefConv $\rightarrow$ Conv | 46.92 | 18.08 |
| | TPV $\rightarrow$ ResBlock | 47.02 | 18.32 |
| | Task-decoupled $\rightarrow$ Task-shared | 46.81 | 17.88 |
| | Full model | **47.36** | **18.73** |

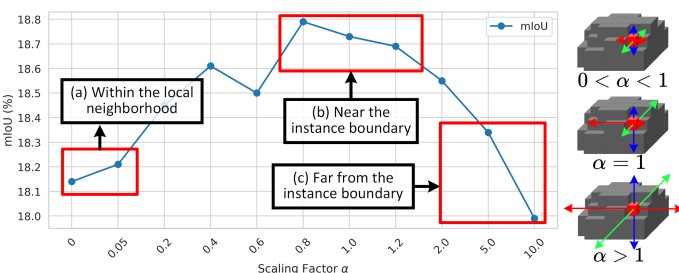

Figure 5: Justifying the aggregation design using voxels at regressed positions $\mathbf{V}_\delta$. By modulating regressed offsets $\Delta$ with a scaling factor $\alpha \in \mathbb{R}$, for each voxel $\mathbf{V}_{i,j,k}$, we select the voxels $\mathbf{V}_\delta$ based on modulated offsets $\Delta = \alpha \Delta_{i,j,k}$, where increasing $\alpha$ will select voxels at larger distances and vice versa.

existing camera-based methods. *Notably, VoxDet achieves the best on all instance-related classes, showing an impressive capacity in understanding outdoor agents.*

## 5.2 Quantitative Study

**Extension to LiDAR-based SSC.** In Tab. 3, we deploy VoxDet with LiDAR input and compare it with the hidden test set of SemanticKITTI. Our VoxDet-L achieves the new record with a 63.0 IoU and 26.0 mIoU, significantly surpassing state-of-the-art methods [60, 57]. Our method achieves the best on large-scale objects like *building, vegetation*, middle-scale objects like *car*, and small-scale objects such as *pole, traffic-sign*. Notably, with our effective regression-include SSC formulation, VoxDet-L achieves a significantly higher IoU of 63.0 without using extra labels/data/temporal information/models, ranking **1st on the CodaLab leaderboard** (until May 22). This capability is crucial for autonomous navigation to prevent collisions with obstacles.

**Ablation Study.** Tab. 4 presents the ablation study, highlighting the following insightful observations. *Line (b):* Compared with the baseline, introducing the REG branch in TDP gives a significant 2.71 IoU gains owing to the better instance-level perception. Interestingly, the limited 0.14 mIoU shows that the REG does not contribute to the semantic discrimability, aligning with our motivation. ***Line (c):*** After deploying the CLS with instance-level aggregation, a significant 1.43 mIoU gain can be

Table 6: Efficiency comparison with SoTA methods.

| Method | $N_{param} \downarrow$ | $T_{inf} \downarrow$ | IoU (%) ↑ | mIoU (%) ↑ |
|---|---|---|---|---|
| OccFormer [82][ICCV'23] | 214 | 199 | 36.42 | 13.50 |
| StereoScene [25][IJCAI'24] | 117 | 258 | 43.85 | 15.43 |
| CGFormer [79][NeurIPS'24] | 122 | 205 | 45.99 | 16.89 |
| SGFormer [18][CVPR'25] | 126 | - | 45.01 | 16.68 |
| ScanSSC [2][CVPR'25] | 145 | 261 | 45.95 | 17.12 |
| **VoxDet (Ours)** | **53** | **159** | **47.36** | **18.73** |

Table 7: Results with monocular depth.

| Method | IoU (%) | mIoU (%) |
|---|---|---|
| VoxFormer-S [34] | 38.68 | 10.67 |
| VoxFormer-T [34] | 38.08 | 11.27 |
| Symphonize [22] | 38.37 | 12.20 |
| OccFormer [82] | 36.50 | 13.46 |
| CGFormer [79] | 41.82 | 14.06 |
| **VoxDet (Ours)** | **43.92** | **16.35** |

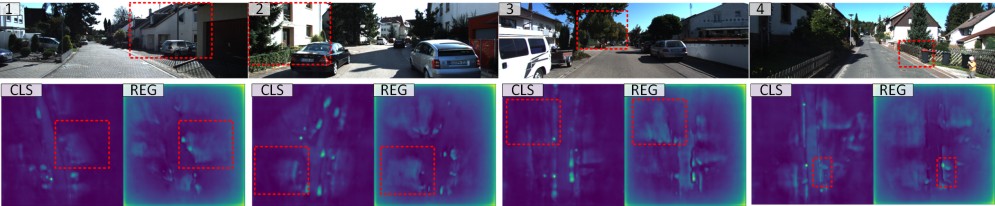

Figure 6: Visualization of the decoupled feature for classification (CLS) and regression (REG).

observed, revealing the importance of instance guidance in voxel prediction. ***Line (d-f):*** By gradually decoupling the volume, both metrics improve progressively, eliminating the task misalignment.

**Different SSC Formulations.** We provide a detailed analysis of each SSC formulation and make a comparison in Tab. 8. *(a) Segmentation (previous SSC works)*: Each voxel only predicts its class. This paradigm uniformly treats all voxels of the same class as a single class group and is entirely agnostic to instance-level geometry,

|  | (a) Seg. | (b) FCOS Det. | (c) VoxDet. |
|---|---|---|---|
| IoU | 42.71 | 45.42 | 47.36 |
| mIoU | 16.28 | 16.42 | 18.73 |

Table 8: Different SSC formulations.

such as length. Hence, it risks incorrectly estimating object instance scales, such as extremely long cars. *(b) Naive FCOS-based detection*: Each voxel regresses the borders of objects and predicts the class simultaneously [56]. By capturing instance-level geometry, this can use the regressed object instances to represent the scene with multiple instance groups. *(c) Our VoxDet*: Each voxel first regresses the borders of object instances, aggregates instance-level context, and predicts the class. Compared with (b), we use regressed offsets to enhance class prediction with instance aggregation.

**Design Variations.** In Tab. 5, we delve into each module with different design variants. For **TDP**, replacing the $\Delta$-guided instance-level aggregation (Eq. 6) with learned deformable self-attention and soft-weights both decreases the performance, revealing the necessity of explicit instance guidance. *The $\Delta$ guidance and attentive design may relieve the negative influence of regression outliers far from instances, consistent with Fig. 5.* In **SVE**, we observe a consistent decline by replacing dense projection, DefConv, TPV, and decoupling with average pooling, Conv, ResBlock, and task-shared designs, respectively. This can verify that our spatial task-decoupling in the TPV space is optimal.

**Model Efficiency.** In Tab. 6, we compare with state-of-the-art works in parameters ($N_{param}$) and inference time ($T_{inf}$). VoxDet uses significantly fewer parameters (M) and less inference time (ms), setting new records on all IoU metrics, highlighting the effectiveness of our new SSC formulation.

**Comparison with Monocular Depth.** In Tab. 7, we make a comparison with state-of-the-art methods by using monocular depth [5]. Our VoxDet also achieves the best results on both metrics, surpassing the previous best entry with 2.10 IoU and 2.29 mIoU, showing significant robustness on depth.

**Exploring Instance-level Aggregation.** Fig. 5 carefully analyzes our aggregation designs (Eq. 6). Instead of directly aggregating regressed voxels $\mathbf{V}_\delta$ with $\Delta$, we modulate it by a scaling factor $\alpha \in \mathbb{R}$, and aggregate at the scaled offset $\tilde{\Delta} = \alpha\Delta$, revealing three observations. ***(1) Local aggregation does not work ($\alpha \to 0$).*** When $\alpha$ approaches zero, performance degrades noticeably. Naively aggregating neighborhoods yields no benefit, as it degenerates to convolutional kernels with local receptive fields. ***(2) Near-boundary aggregation helps ($\alpha \approx 1$).*** As $\alpha$ grows toward 1.0, performance steadily improves, indicating that the boundary contributes more informative signals for dense perception. Notably, $\alpha = 0.8$ slightly outperforms our default of $\alpha = 1.0$, likely by suppressing outliers outside instances. ***(3) Outside-instance aggregation hurts ($\alpha > 1$).*** When $\alpha$ exceeds 1.0 substantially, performance declines sharply, revealing the negative impact of aggregating voxels beyond the instance extent.

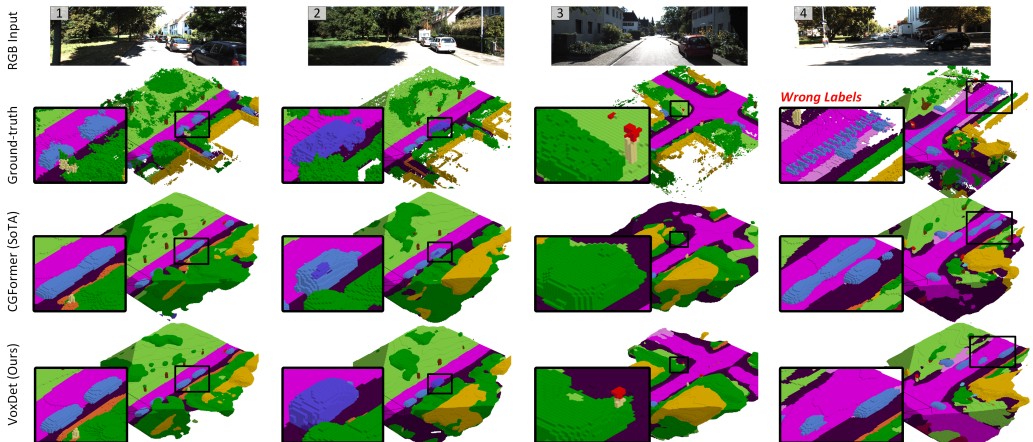

Figure 7: Qualitative comparison with state-of-the-art method [79]. Zoom in for a better view.

## 5.3 Qualitative Results

**Disentangled Volume.** In Fig. 6, we visualize the decoupled feature volume (Eq. 3) at Bird's Eye View (BEV) view for the classification (CLS) and regression (REG), revealing the following three insightful observations. **(1)** Different from CLS, focusing on semantically informative local regions, REG naturally seeks to discover the more complete boundary of instances. This capacity is critical in SSC, as completing the scene is the key focus. **(2)** Although REG has not been trained with semantic supervision, it can discover informative cues of potential objects (4th sample) with orthogonal effects with CLS. **(3)** We can see the four borders of the BEV map are activated, as the 3D volume borders are unified boundaries of adjacent large-scale instances, like vegetation and buildings.

**Semantic Scene Completion.** Fig. 7 visually compares our VoxDet with state-of-the-art method [79]. VoxDet shows noticeable gains in geometric completion, such as complete *car* borders sample 1, better infers instance-level semantics, such as the *truck* in sample 2, and can correctly detect the *challenging yet important traffic sign* (3rd sample). This is mainly owing to our instance-centric formulation, which enhances spatial perception and semantic understanding of object instances. Notably, in the 4th sample, VoxDet has built-in safeguards (size-based filtering) that mitigate the wrong-label concern, making SSC supervision more informative and safer (see Appx. B).

## 6 Conclusion

In this work, we first reveal an ever-overlooked free lunch in SSC labels: the voxel-level class label has implicitly provided the instance-level insight thanks to its occlusion-free nature. Then, we propose **VoxNT**, a simple yet effective algorithm that freely converts class labels to the instance-level offsets. Based on this, we propose **VoxDet**, a new detection-based formulation for instance-centric SSC. VoxDet first spatially decouples 3D volumes to generate task-specific features, learning spatial deformations in the tri-perceptive space. Next, it adopts a task-decoupled predictor to generate instance-aware predictions guided by the regressed 4D offset field. Extensive experiments reveal a lot of insightful phenomena for the following works and verify the state-of-the-art role of VoxDet.

## 7 Discussion

**Safety Clarification.** While our method effectively discovers informative geometric cues for naturally separated instances (e.g., traffic signs, poles, and trunks), we acknowledge that our assumption may not hold in crowded or highly dense scenes. In cases where instances are naturally merged in ground-truth labels (such as overlapping tree crowns), our method may fall back to conventional SSC methods by treating these as a single class, without instance-level distinction. This may limit our performance gains in complex perception scenarios and could affect downstream tasks that heavily rely on precise object separation. For such challenging cases, we recommend incorporating more reliable instance-level labels (either from human annotation or large foundational models) to provide more accurate supervision and further enhance the safety and reliability of downstream applications.

## Acknowledgment

We would like to express our gratitude to Wentao Pan, Reyhaneh Hosseininejad, Yasamin Borhani, Yuanfan Zheng, and Hengyu Liu for their insightful discussions, which contributed to the improvement of this work. Additionally, we gratefully acknowledge the financial support provided by Valeo.

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

# NeurIPS Paper Checklist

1. **Claims**

   Question: Do the main claims made in the abstract and introduction accurately reflect the paper's contributions and scope?

   Answer: [Yes]

   Justification: The abstract and introduction sections offer a comprehensive discussion of the context, intuition, insight, ambitions, and contributions of the manuscript.

   Guidelines:

   - The answer NA means that the abstract and introduction do not include the claims made in the paper.
   - The abstract and/or introduction should clearly state the claims made, including the contributions made in the paper and important assumptions and limitations. A No or NA answer to this question will not be perceived well by the reviewers.
   - The claims made should match theoretical and experimental results, and reflect how much the results can be expected to generalize to other settings.
   - It is fine to include aspirational goals as motivation as long as it is clear that these goals are not attained by the paper.

2. **Limitations**

   Question: Does the paper discuss the limitations of the work performed by the authors?

   Answer: [Yes]

   Justification: Yes. The limitations of this work have been comprehensively discussed in the Appendix due to the page limit of the main paper.

   Guidelines:

   - The answer NA means that the paper has no limitation while the answer No means that the paper has limitations, but those are not discussed in the paper.
   - The authors are encouraged to create a separate "Limitations" section in their paper.
   - The paper should point out any strong assumptions and how robust the results are to violations of these assumptions (e.g., independence assumptions, noiseless settings, model well-specification, asymptotic approximations only holding locally). The authors should reflect on how these assumptions might be violated in practice and what the implications would be.
   - The authors should reflect on the scope of the claims made, e.g., if the approach was only tested on a few datasets or with a few runs. In general, empirical results often depend on implicit assumptions, which should be articulated.
   - The authors should reflect on the factors that influence the performance of the approach. For example, a facial recognition algorithm may perform poorly when image resolution is low or images are taken in low lighting. Or a speech-to-text system might not be used reliably to provide closed captions for online lectures because it fails to handle technical jargon.
   - The authors should discuss the computational efficiency of the proposed algorithms and how they scale with dataset size.
   - If applicable, the authors should discuss possible limitations of their approach to address problems of privacy and fairness.
   - While the authors might fear that complete honesty about limitations might be used by reviewers as grounds for rejection, a worse outcome might be that reviewers discover limitations that aren't acknowledged in the paper. The authors should use their best judgment and recognize that individual actions in favor of transparency play an important role in developing norms that preserve the integrity of the community. Reviewers will be specifically instructed to not penalize honesty concerning limitations.

3. **Theory assumptions and proofs**

   Question: For each theoretical result, does the paper provide the full set of assumptions and a complete (and correct) proof?

Answer: [Yes]

Justification: For each theoretical result, the paper provides the full set of assumptions and a complete (and correct) justification.

Guidelines:

- The answer NA means that the paper does not include theoretical results.
- All the theorems, formulas, and proofs in the paper should be numbered and cross-referenced.
- All assumptions should be clearly stated or referenced in the statement of any theorems.
- The proofs can either appear in the main paper or the supplemental material, but if they appear in the supplemental material, the authors are encouraged to provide a short proof sketch to provide intuition.
- Inversely, any informal proof provided in the core of the paper should be complemented by formal proofs provided in appendix or supplemental material.
- Theorems and Lemmas that the proof relies upon should be properly referenced.

4. **Experimental result reproducibility**

Question: Does the paper fully disclose all the information needed to reproduce the main experimental results of the paper to the extent that it affects the main claims and/or conclusions of the paper (regardless of whether the code and data are provided or not)?

Answer: [Yes]

Justification: In the Appendix, we have presented the implementation details and the experimental setups with corresponding reproducible credentials. Besides, we have submitted the raw evaluation results from the CodaLab online platform of the hidden test set. We ensure the code and models will be open-source.

Guidelines:

- The answer NA means that the paper does not include experiments.
- If the paper includes experiments, a No answer to this question will not be perceived well by the reviewers: Making the paper reproducible is important, regardless of whether the code and data are provided or not.
- If the contribution is a dataset and/or model, the authors should describe the steps taken to make their results reproducible or verifiable.
- Depending on the contribution, reproducibility can be accomplished in various ways. For example, if the contribution is a novel architecture, describing the architecture fully might suffice, or if the contribution is a specific model and empirical evaluation, it may be necessary to either make it possible for others to replicate the model with the same dataset, or provide access to the model. In general. releasing code and data is often one good way to accomplish this, but reproducibility can also be provided via detailed instructions for how to replicate the results, access to a hosted model (e.g., in the case of a large language model), releasing of a model checkpoint, or other means that are appropriate to the research performed.
- While NeurIPS does not require releasing code, the conference does require all submissions to provide some reasonable avenue for reproducibility, which may depend on the nature of the contribution. For example
  - (a) If the contribution is primarily a new algorithm, the paper should make it clear how to reproduce that algorithm.
  - (b) If the contribution is primarily a new model architecture, the paper should describe the architecture clearly and fully.
  - (c) If the contribution is a new model (e.g., a large language model), then there should either be a way to access this model for reproducing the results or a way to reproduce the model (e.g., with an open-source dataset or instructions for how to construct the dataset).
  - (d) We recognize that reproducibility may be tricky in some cases, in which case authors are welcome to describe the particular way they provide for reproducibility. In the case of closed-source models, it may be that access to the model is limited in some way (e.g., to registered users), but it should be possible for other researchers to have some path to reproducing or verifying the results.

5. **Open access to data and code**

   Question: Does the paper provide open access to the data and code, with sufficient instructions to faithfully reproduce the main experimental results, as described in supplemental material?

   Answer: [Yes]

   Justification: All data used in this study are drawn from publicly accessible platforms and employed following existing works. The cleaned codebase and trained models, including a comprehensive README, have been released publicly.

   Guidelines:

   - The answer NA means that paper does not include experiments requiring code.
   - Please see the NeurIPS code and data submission guidelines (`https://nips.cc/public/guides/CodeSubmissionPolicy`) for more details.
   - While we encourage the release of code and data, we understand that this might not be possible, so "No" is an acceptable answer. Papers cannot be rejected simply for not including code, unless this is central to the contribution (e.g., for a new open-source benchmark).
   - The instructions should contain the exact command and environment needed to run to reproduce the results. See the NeurIPS code and data submission guidelines (`https://nips.cc/public/guides/CodeSubmissionPolicy`) for more details.
   - The authors should provide instructions on data access and preparation, including how to access the raw data, preprocessed data, intermediate data, and generated data, etc.
   - The authors should provide scripts to reproduce all experimental results for the new proposed method and baselines. If only a subset of experiments are reproducible, they should state which ones are omitted from the script and why.
   - At submission time, to preserve anonymity, the authors should release anonymized versions (if applicable).
   - Providing as much information as possible in supplemental material (appended to the paper) is recommended, but including URLs to data and code is permitted.

6. **Experimental setting/details**

   Question: Does the paper specify all the training and test details (e.g., data splits, hyperparameters, how they were chosen, type of optimizer, etc.) necessary to understand the results?

   Answer: [Yes]

   Justification: In the Appendix, we have presented the implementation and experimental details with corresponding reproducible credentials. Besides, we have submitted the raw evaluation results from the CodaLab online platform of the hidden test set.

   Guidelines:

   - The answer NA means that the paper does not include experiments.
   - The experimental setting should be presented in the core of the paper to a level of detail that is necessary to appreciate the results and make sense of them.
   - The full details can be provided either with the code, in appendix, or as supplemental material.

7. **Experiment statistical significance**

   Question: Does the paper report error bars suitably and correctly defined or other appropriate information about the statistical significance of the experiments?

   Answer: [Yes]

   Justification: The results contain the standard deviation of the results over several random runs. We have presented the experiments in the Appendix.

   Guidelines:

   - The answer NA means that the paper does not include experiments.

- The authors should answer "Yes" if the results are accompanied by error bars, confidence intervals, or statistical significance tests, at least for the experiments that support the main claims of the paper.
- The factors of variability that the error bars are capturing should be clearly stated (for example, train/test split, initialization, random drawing of some parameter, or overall run with given experimental conditions).
- The method for calculating the error bars should be explained (closed form formula, call to a library function, bootstrap, etc.)
- The assumptions made should be given (e.g., Normally distributed errors).
- It should be clear whether the error bar is the standard deviation or the standard error of the mean.
- It is OK to report 1-sigma error bars, but one should state it. The authors should preferably report a 2-sigma error bar than state that they have a 96% CI, if the hypothesis of Normality of errors is not verified.
- For asymmetric distributions, the authors should be careful not to show in tables or figures symmetric error bars that would yield results that are out of range (e.g. negative error rates).
- If error bars are reported in tables or plots, The authors should explain in the text how they were calculated and reference the corresponding figures or tables in the text.

8. **Experiments compute resources**

Question: For each experiment, does the paper provide sufficient information on the computer resources (type of compute workers, memory, time of execution) needed to reproduce the experiments?

Answer: [Yes]

Justification: The details of experiments are presented with corresponding reproducible credentials.

Guidelines:

- The answer NA means that the paper does not include experiments.
- The paper should indicate the type of compute workers CPU or GPU, internal cluster, or cloud provider, including relevant memory and storage.
- The paper should provide the amount of compute required for each of the individual experimental runs as well as estimate the total compute.
- The paper should disclose whether the full research project required more compute than the experiments reported in the paper (e.g., preliminary or failed experiments that didn't make it into the paper).

9. **Code of ethics**

Question: Does the research conducted in the paper conform, in every respect, with the NeurIPS Code of Ethics https://neurips.cc/public/EthicsGuidelines?

Answer: [Yes]

Justification: The research conducted in the paper conforms with the NeurIPS Code of Ethics.

Guidelines:

- The answer NA means that the authors have not reviewed the NeurIPS Code of Ethics.
- If the authors answer No, they should explain the special circumstances that require a deviation from the Code of Ethics.
- The authors should make sure to preserve anonymity (e.g., if there is a special consideration due to laws or regulations in their jurisdiction).

10. **Broader impacts**

Question: Does the paper discuss both potential positive societal impacts and negative societal impacts of the work performed?

Answer: [Yes]

Justification: The paper has discussed the border impacts in the Appendix.

Guidelines:

- The answer NA means that there is no societal impact of the work performed.
- If the authors answer NA or No, they should explain why their work has no societal impact or why the paper does not address societal impact.
- Examples of negative societal impacts include potential malicious or unintended uses (e.g., disinformation, generating fake profiles, surveillance), fairness considerations (e.g., deployment of technologies that could make decisions that unfairly impact specific groups), privacy considerations, and security considerations.
- The conference expects that many papers will be foundational research and not tied to particular applications, let alone deployments. However, if there is a direct path to any negative applications, the authors should point it out. For example, it is legitimate to point out that an improvement in the quality of generative models could be used to generate deepfakes for disinformation. On the other hand, it is not needed to point out that a generic algorithm for optimizing neural networks could enable people to train models that generate Deepfakes faster.
- The authors should consider possible harms that could arise when the technology is being used as intended and functioning correctly, harms that could arise when the technology is being used as intended but gives incorrect results, and harms following from (intentional or unintentional) misuse of the technology.
- If there are negative societal impacts, the authors could also discuss possible mitigation strategies (e.g., gated release of models, providing defenses in addition to attacks, mechanisms for monitoring misuse, mechanisms to monitor how a system learns from feedback over time, improving the efficiency and accessibility of ML).

11. **Safeguards**

Question: Does the paper describe safeguards that have been put in place for responsible release of data or models that have a high risk for misuse (e.g., pretrained language models, image generators, or scraped datasets)?

Answer: [NA]

Justification: This study does not involve any generative models that pose a risk of misuse.

Guidelines:

- The answer NA means that the paper poses no such risks.
- Released models that have a high risk for misuse or dual-use should be released with necessary safeguards to allow for controlled use of the model, for example by requiring that users adhere to usage guidelines or restrictions to access the model or implementing safety filters.
- Datasets that have been scraped from the Internet could pose safety risks. The authors should describe how they avoided releasing unsafe images.
- We recognize that providing effective safeguards is challenging, and many papers do not require this, but we encourage authors to take this into account and make a best faith effort.

12. **Licenses for existing assets**

Question: Are the creators or original owners of assets (e.g., code, data, models), used in the paper, properly credited and are the license and terms of use explicitly mentioned and properly respected?

Answer: [Yes]

Justification: All external assets, including data, code, and models, are credited to their original owners, with their licenses and usage terms clearly specified and fully honored.

Guidelines:

- The answer NA means that the paper does not use existing assets.
- The authors should cite the original paper that produced the code package or dataset.
- The authors should state which version of the asset is used and, if possible, include a URL.
- The name of the license (e.g., CC-BY 4.0) should be included for each asset.

- For scraped data from a particular source (e.g., website), the copyright and terms of service of that source should be provided.
- If assets are released, the license, copyright information, and terms of use in the package should be provided. For popular datasets, `paperswithcode.com/datasets` has curated licenses for some datasets. Their licensing guide can help determine the license of a dataset.
- For existing datasets that are re-packaged, both the original license and the license of the derived asset (if it has changed) should be provided.
- If this information is not available online, the authors are encouraged to reach out to the asset's creators.

13. **New assets**

    Question: Are new assets introduced in the paper well documented and is the documentation provided alongside the assets?

    Answer: [Yes]

    Justification: All newly introduced assets are comprehensively documented and supplied alongside the existing resources.

    Guidelines:

    - The answer NA means that the paper does not release new assets.
    - Researchers should communicate the details of the dataset/code/model as part of their submissions via structured templates. This includes details about training, license, limitations, etc.
    - The paper should discuss whether and how consent was obtained from people whose asset is used.
    - At submission time, remember to anonymize your assets (if applicable). You can either create an anonymized URL or include an anonymized zip file.

14. **Crowdsourcing and research with human subjects**

    Question: For crowdsourcing experiments and research with human subjects, does the paper include the full text of instructions given to participants and screenshots, if applicable, as well as details about compensation (if any)?

    Answer: [NA]

    Justification: This study does not include human subjects.

    Guidelines:

    - The answer NA means that the paper does not involve crowdsourcing nor research with human subjects.
    - Including this information in the supplemental material is fine, but if the main contribution of the paper involves human subjects, then as much detail as possible should be included in the main paper.
    - According to the NeurIPS Code of Ethics, workers involved in data collection, curation, or other labor should be paid at least the minimum wage in the country of the data collector.

15. **Institutional review board (IRB) approvals or equivalent for research with human subjects**

    Question: Does the paper describe potential risks incurred by study participants, whether such risks were disclosed to the subjects, and whether Institutional Review Board (IRB) approvals (or an equivalent approval/review based on the requirements of your country or institution) were obtained?

    Answer: [NA]

    Justification: This paper does not involve crowdsourcing nor research with human subjects.

    Guidelines:

    - The answer NA means that the paper does not involve crowdsourcing nor research with human subjects.

- Depending on the country in which research is conducted, IRB approval (or equivalent) may be required for any human subjects research. If you obtained IRB approval, you should clearly state this in the paper.
- We recognize that the procedures for this may vary significantly between institutions and locations, and we expect authors to adhere to the NeurIPS Code of Ethics and the guidelines for their institution.
- For initial submissions, do not include any information that would break anonymity (if applicable), such as the institution conducting the review.

16. **Declaration of LLM usage**

Question: Does the paper describe the usage of LLMs if it is an important, original, or non-standard component of the core methods in this research? Note that if the LLM is used only for writing, editing, or formatting purposes and does not impact the core methodology, scientific rigorousness, or originality of the research, declaration is not required.

Answer: [NA]

Justification: The core method development in this research does not involve LLMs as any important, original, or non-standard components

Guidelines:

- The answer NA means that the core method development in this research does not involve LLMs as any important, original, or non-standard components.
- Please refer to our LLM policy (`https://neurips.cc/Conferences/2025/LLM`) for what should or should not be described.

This Appendix aims to clarify novelty, practical applications, border impacts, experimental justifications, and future directions, which are organized as follows.

Appendix A: Additional quantitative results
- More comprehensive comparison of camera-based SSC
- Robust analysis with multiple runs
  - Qualified as a new, powerful, robust, lightweight, and efficient baseline for SSC
- Results on SemanticKITTI validation set
- How to define object instances
  - A generalized definition works best, which is different from the 2D intuition
- Sensitivity analysis on hyperparameters
- More comparison of model efficiency

Appendix B: More insights and practical usages of VoxNT trick
- Freely understand instance scales
- Ability to freely identify wrong labels
- Freely eliminate wrong labels in training
- Rethink the evaluation on dynamic objects

Appendix C: Detailed experimental setups
- Datasets and evaluation metrics
- Implementation details
- Algorithmic details of the VoxNT trick

Appendix D: Discussion
- Difference with detection-assisted SSC works
- Broader impacts
- Limitations and future works
- Ethical claims

Appendix E: Additional qualitative results
- Failure case analysis
- More qualitative comparison

# A  Additional Quantitative Results

## A.1  More Comparison of Camera-based SSC

In Tab. 9, we make a more comprehensive comparison on the SemanticKITTI hidden test set. For fair comparison, we also report the performance using the same weight (3.0) of cross-entropy loss with the latest concurrent work [60] replacing our default (1.0), denoted as VoxDet$^{\dagger}$. Based on ResNet-50, our method surpasses all the methods using a more advanced EfficientNet backbone. Additionally, our method achieves at least the best or the second best in all classes, showing superior effectiveness.

## A.2  Robustness Analysis of Multiple Runs

In Fig. 8, we report the results of multiple runs (5 times independent experiments) to justify the robustness of our VoxDet. The curves summarize the IoU and mIoU results for each epoch on the SemanticKITTI validation set. To better illustrate the difference, we also demonstrate the results of the previous state-of-the-art method, CGFormer [79], using the officially released training log.

It can be observed that our VoxDet achieves impressive robustness with multiple runs, giving very similar performance on the IoU and mIoU in different runs. Note that there is some tradeoff between IoU and mIoU metrics, i.e., some runs achieve slightly higher IoU while sacrificing a little mIoU. Additionally, our method achieves visually significant gains over CGFormer on both the robustness and IoU/mIoU metrics, highlighting the strength VoxDet. ***Hence, due to the superior effectiveness, robustness, and efficiency, we believe that VoxDet is qualified to serve as a powerful, lightweight, and efficient baseline model for the following works.***

Table 9: Results of Camera-based SSC on SemanticKITTI [3] hidden test set. **T** is using extra temporal information. The best and the second best results are in **bold** and underlined, respectively. R-50 indicates ResNet-50, and Eff-B7 is the stronger EfficientNet-B7 backbone with more parameters. † indicates setting the same weight (3.0) on the cross-entropy loss as the recent work [60].

| Method | Arch. | T | IoU | mIoU | road | sidewalk | parking | other-grnd. | building | car | truck | bicycle | motorcycle | other-veh. | vegetation | trunk | terrain | person | bicyclist | motorcyclist | fence | pole | traf.-sign |
|---|---|---|---|---|---|---|---|---|---|---|---|---|---|---|---|---|---|---|---|---|---|---|---|
| MonoScene* [6] | Eff-B7 | | 34.16 | 11.08 | 54.70 | 27.10 | 24.80 | 5.70 | 14.40 | 18.80 | 3.30 | 0.50 | 0.70 | 4.40 | 14.90 | 2.40 | 19.50 | 1.00 | 1.40 | 0.40 | 11.10 | 3.30 | 2.10 |
| TPVFormer [21] | Eff-B7 | | 34.25 | 11.26 | 55.10 | 27.20 | 27.40 | 6.50 | 14.80 | 19.20 | 3.70 | 1.00 | 0.50 | 2.30 | 13.90 | 2.60 | 20.40 | 1.10 | 2.40 | 0.30 | 11.00 | 2.90 | 1.50 |
| SurroundOcc [65] | Eff-B7 | | 34.72 | 11.86 | 56.90 | 28.30 | 30.20 | 6.80 | 15.20 | 20.60 | 1.40 | 1.60 | 1.20 | 4.40 | 14.90 | 3.40 | 19.30 | 1.40 | 2.00 | 0.10 | 11.30 | 3.90 | 2.40 |
| OccFormer [82] | Eff-B7 | | 34.53 | 12.32 | 55.90 | 30.30 | 31.50 | 6.50 | 15.70 | 21.60 | 1.20 | 1.50 | 1.70 | 3.20 | 16.80 | 3.90 | 21.30 | 2.20 | 1.10 | 0.20 | 11.90 | 3.80 | 3.70 |
| IAMSSC [67] | R-50 | | 43.74 | 12.37 | 54.00 | 25.50 | 24.70 | 6.90 | 19.20 | 21.30 | 3.80 | 1.10 | 0.60 | 3.90 | 22.70 | 5.80 | 19.40 | 1.50 | 2.90 | 0.50 | 11.90 | 5.30 | 4.10 |
| VoxFormer [34] | R-50 | | 42.95 | 12.20 | 53.90 | 25.30 | 21.10 | 5.60 | 19.80 | 20.80 | 3.50 | 1.00 | 0.70 | 3.70 | 22.40 | 7.50 | 21.30 | 1.40 | 2.60 | 0.20 | 11.10 | 5.10 | 4.90 |
| VoxFormer [34] | R-50 | ✓ | 43.21 | 13.41 | 54.10 | 26.90 | 25.10 | 7.30 | 23.50 | 21.70 | 3.60 | 1.90 | 1.60 | 4.10 | 24.40 | 8.10 | 24.20 | 1.60 | 1.10 | 0.00 | 13.10 | 6.60 | 5.70 |
| DepthSSC [76] | R-50 | | 44.58 | 13.11 | 55.64 | 27.25 | 25.72 | 5.78 | 20.46 | 21.94 | 3.74 | 1.35 | 0.98 | 4.17 | 23.37 | 7.64 | 21.56 | 1.34 | 2.79 | 0.28 | 12.94 | 5.87 | 6.23 |
| Symphonize [22] | R-50 | | 42.19 | 15.04 | 58.40 | 29.30 | 26.90 | 11.70 | 24.70 | 23.60 | 3.20 | 3.60 | 2.60 | 5.60 | 24.20 | 10.00 | 23.10 | **3.20** | 1.90 | **2.00** | 16.10 | 7.70 | 8.00 |
| HASSC [61] | R-50 | | 43.40 | 13.34 | 54.60 | 27.70 | 23.80 | 6.20 | 21.10 | 22.80 | 4.70 | 1.60 | 1.00 | 3.90 | 23.80 | 8.50 | 23.30 | 1.60 | 4.00 | 0.30 | 13.10 | 5.80 | 5.50 |
| HASSC [61] | R-50 | ✓ | 42.87 | 14.38 | 55.30 | 29.60 | 25.90 | 11.30 | 23.10 | 23.00 | 2.90 | 1.90 | 1.50 | 4.90 | 24.80 | 9.80 | 26.50 | 1.40 | 3.00 | 0.00 | 14.30 | 7.00 | 7.10 |
| StereoScene [25] | Eff-B7 | | 43.34 | 15.36 | 61.90 | 31.20 | 30.70 | 10.70 | 24.20 | 22.80 | 2.80 | 3.40 | 2.40 | **6.10** | 23.80 | 8.40 | 27.00 | 2.90 | 2.20 | 0.50 | 16.50 | 7.00 | 7.20 |
| H2GFormer [64] | R-50 | | 44.20 | 13.72 | 56.40 | 28.60 | 26.50 | 4.90 | 22.80 | 23.40 | 4.80 | 0.80 | 0.90 | 4.10 | 24.60 | 9.10 | 23.80 | 1.20 | 2.50 | 0.10 | 13.30 | 6.40 | 6.30 |
| H2GFormer [64] | R-50 | ✓ | 43.52 | 14.60 | 57.90 | 30.40 | 30.00 | 6.90 | 24.00 | 23.70 | 5.20 | 0.60 | 1.20 | 5.00 | 25.20 | 10.70 | 25.80 | 1.10 | 0.10 | 0.00 | 14.60 | 7.50 | 9.30 |
| MonoOcc [83] | R-50 | | - | 13.80 | 55.20 | 27.80 | 25.10 | 9.70 | 21.40 | 23.20 | 5.20 | 2.20 | 1.50 | 5.40 | 24.00 | 8.70 | 23.00 | 1.70 | 2.00 | 0.20 | 13.40 | 5.80 | 6.40 |
| CGFormer [79] | Eff-B7 | | 44.41 | 16.63 | 64.30 | 34.20 | 34.10 | 12.10 | 25.80 | 26.10 | 4.30 | 3.70 | 1.30 | 2.70 | 24.50 | 11.20 | 29.30 | 1.70 | 3.60 | 0.40 | 18.70 | 8.70 | 9.30 |
| L2COcc-C [60] | Eff-B7 | | 44.31 | 17.03 | **66.00** | 35.00 | 33.10 | 13.50 | 25.10 | 27.20 | 3.00 | 3.50 | 3.60 | 4.30 | 25.20 | 11.50 | 30.10 | 1.50 | 2.40 | 0.20 | 20.50 | 9.10 | 8.90 |
| HTCL [24] | Eff-B7 | ✓ | 44.23 | 17.09 | 64.40 | 34.80 | 33.80 | 12.40 | 25.90 | **27.30** | 5.70 | 1.80 | 2.20 | 5.40 | 25.30 | 10.80 | 31.20 | 1.10 | 3.10 | 0.90 | 21.10 | 9.00 | 8.30 |
| **VoxDet (Ours)** | R-50 | | 47.27 | 18.47 | 64.70 | 35.50 | 34.80 | **14.40** | 28.10 | 26.90 | **6.10** | **5.90** | 5.10 | 5.00 | 28.70 | **13.60** | 31.70 | 3.10 | **4.00** | 1.30 | 21.50 | **10.10** | 10.30 |
| **VoxDet† (Ours)** | R-50 | | **47.81** | **18.67** | 65.50 | **36.10** | **35.50** | 13.20 | **28.40** | 27.30 | 5.40 | 4.60 | **5.40** | 5.40 | **29.50** | 13.10 | **32.00** | 3.10 | **6.10** | 0.90 | **22.10** | 10.20 | **11.10** |

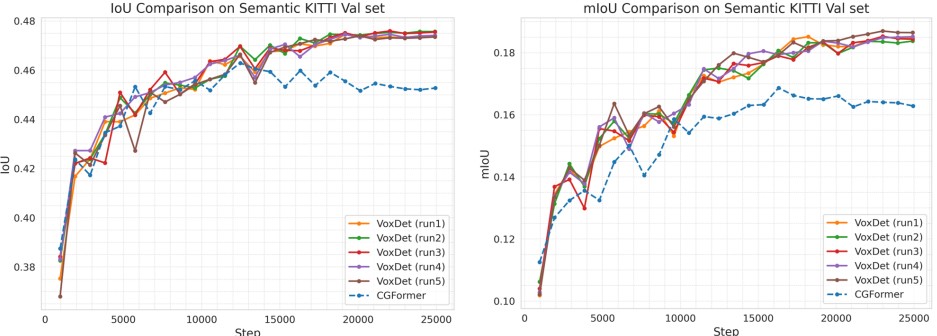

Figure 8: Robustness analysis of our VoxDet with multiple runs. We report the per-epoch validation results. The results from the official training log (not our reproduced results) given by the previous state-of-the-art method, CGFomrer [79], are also listed for comparison. We can observe visually significant improvements in both performance and robustness of our method.

## A.3 How to Define Object Instances?

In Tab. 10, we explore different definitions of objects by removing the instance-level regression for specific classes, including (1) background, including road, sidewalk, etc (2) empty class. We empirically find that a generalized definition of objects, i.e., all the semantic classes, performs best. This is different from the intuition in the 2D image domain, which only considers objects with well-defined shapes.

Table 10: Analysis of object definitions by removing instance-level regression for specific classes.

| Setting | IoU | mIoU |
|---|---|---|
| w/o. regress background | 46.78 | 18.19 |
| w/o. regress empty | 47.08 | 18.05 |
| Full model | **47.36** | **18.73** |

## A.4 Results on SemanticKITTI Validation Set

Tab. 11 presents the comparison on the SemanticKITTI validation set. Our VoxDet achieves the best results of 47.36 IoU and 18.73 mIoU, surpassing the second-best counterpart [24] with a noticeable 1.60 mIoU gains. Compared with the previous state-of-the-art method [79] with 45.99 IoU and 16.87 mIoU, our VoxDet gives a 1.37 IoU and 1.86 mIoU gains, verifying the superior effectiveness. Besides, VoxDet achieves the best and second-best results in 17 of the 19 classes, which shows its effectiveness. Specifically, our method performs well on object instances, achieving the best on *building, car, truck, motorcycle, traffic sign, etc*, revealing the superior instance-centric learning.

Table 11: Quantitative results on SemanticKITTI [3] validation set. T indicates using extra temporal information. The best and the second best results are in bold and underlined, respectively.

| Method | T | IoU | mIoU | road (15.30%) | sidewalk (11.13%) | parking (1.12%) | other-grnd. (0.56%) | building (14.1%) | car (3.92%) | truck (0.16%) | bicycle (0.03%) | motorcycle (0.03%) | other-veh. (0.20%) | vegetation (39.3%) | trunk (0.51%) | terrain (9.17%) | person (0.07%) | bicyclist (0.07%) | motorcyclist (0.05%) | fence (3.90%) | pole (0.29%) | traf.-sign (0.08%) |
|---|---|---|---|---|---|---|---|---|---|---|---|---|---|---|---|---|---|---|---|---|---|---|
| MonoScene [6] | | 36.86 | 11.08 | 56.52 | 26.72 | 14.27 | 0.46 | 14.09 | 23.26 | 6.98 | 0.61 | 0.45 | 1.48 | 17.89 | 2.81 | 29.64 | 1.86 | 1.20 | 0.00 | 5.84 | 4.14 | 2.25 |
| TPVFormer [21] | | 35.61 | 11.36 | 56.50 | 25.87 | 20.60 | 0.85 | 13.88 | 23.81 | 8.08 | 0.36 | 0.05 | 4.35 | 16.92 | 2.26 | 30.38 | 0.51 | 0.89 | 0.00 | 5.94 | 3.14 | 1.52 |
| OccFormer [82] | | 36.50 | 13.46 | 58.85 | 26.88 | 19.61 | 0.31 | 14.40 | 25.09 | 25.53 | 0.81 | 1.19 | 8.52 | 19.63 | 3.93 | 32.62 | 2.78 | 2.82 | 0.00 | 5.61 | 4.26 | 2.86 |
| IAMSSC [67] | | 44.29 | 12.45 | 54.55 | 25.85 | 16.02 | 0.70 | 17.38 | 26.26 | 8.74 | 0.60 | 0.15 | 5.06 | 24.63 | 4.95 | 30.13 | 1.32 | 3.46 | 0.01 | 6.86 | 6.35 | 3.56 |
| VoxFormer [34] | | 44.02 | 12.35 | 54.76 | 26.35 | 15.50 | 0.70 | 17.65 | 25.79 | 5.63 | 0.59 | 0.51 | 3.77 | 24.39 | 5.08 | 29.96 | 1.78 | 3.32 | 0.00 | 7.64 | 7.11 | 4.18 |
| VoxFormer [34] | ✓ | 44.15 | 13.35 | 53.57 | 26.52 | 19.69 | 0.42 | 19.54 | 26.54 | 7.26 | 1.28 | 0.56 | 7.81 | 26.10 | 6.10 | 33.06 | 1.93 | 1.97 | 0.00 | 7.31 | 9.15 | 4.94 |
| Symphonize [22] | | 41.92 | 14.89 | 56.37 | 27.58 | 15.28 | 0.95 | 21.64 | 28.68 | 20.44 | 2.54 | 2.82 | 13.89 | 25.72 | 6.60 | 30.87 | 3.52 | 2.24 | 0.00 | 8.40 | 9.57 | 5.76 |
| HASSC [61] | | 44.82 | 13.48 | 57.05 | 28.25 | 15.90 | **1.04** | 19.05 | 27.23 | 9.91 | 0.92 | 0.86 | 5.61 | 25.48 | 6.15 | 32.94 | 2.80 | **4.71** | 0.00 | 6.58 | 7.68 | 4.05 |
| H2GFormer [64] | | 44.57 | 13.73 | 56.08 | 29.12 | 17.83 | 0.45 | 19.74 | 28.21 | 10.00 | 0.50 | 0.47 | 7.39 | 26.25 | 6.80 | 34.42 | 1.54 | 2.88 | 0.00 | 7.24 | 7.88 | 4.68 |
| H2GFormer [64] | ✓ | 44.69 | 14.29 | 57.00 | 29.37 | 21.74 | 0.34 | 20.51 | 28.21 | 6.80 | 0.95 | 0.91 | 9.32 | 27.44 | 7.80 | 36.26 | 1.15 | 0.10 | 0.00 | 7.98 | 9.88 | 5.81 |
| SGN [44] | | 43.60 | 14.55 | 59.32 | 30.51 | 18.46 | 0.42 | 21.43 | 31.88 | 13.18 | 0.58 | 0.17 | 5.68 | 25.98 | 7.43 | 34.42 | 1.28 | 1.49 | 0.00 | 9.66 | 9.83 | 4.71 |
| SGN [44] | ✓ | 46.21 | 15.32 | 59.10 | 29.41 | 19.05 | 0.33 | 25.17 | 33.31 | 6.03 | 0.61 | 0.46 | 9.84 | **28.93** | 9.58 | 38.12 | 0.47 | 0.10 | 0.00 | 9.96 | 13.25 | 7.32 |
| CGFormer [79] | | 45.99 | 16.87 | 65.51 | 32.31 | 20.82 | 0.16 | 23.52 | 34.32 | 19.44 | **4.61** | 2.71 | 7.67 | 26.93 | 8.83 | 39.54 | 2.38 | 4.08 | 0.00 | 9.20 | 10.67 | 7.84 |
| HTCL [24] | ✓ | 45.51 | 17.13 | 63.70 | 32.48 | **23.27** | 0.14 | 24.13 | 34.30 | 20.72 | 3.99 | 2.80 | 11.99 | 26.96 | 8.79 | 37.73 | 2.56 | 2.30 | 0.00 | 11.22 | 11.49 | 6.95 |
| **VoxDet (Ours)** | | **47.36** | **18.73** | **65.55** | **34.22** | 20.88 | 0.04 | **25.79** | **34.50** | **31.05** | 3.95 | **5.14** | **14.65** | 28.93 | **10.20** | **41.27** | **4.48** | 3.14 | 0.00 | **11.73** | **12.19** | **8.28** |

Table 12: Analysis on instance-level aggregation layers $N$.

| $N$ | IoU | mIoU |
|---|---|---|
| 1 | 47.06 | 18.20 |
| 2 | **47.38** | 18.32 |
| 3 | 47.34 | 18.54 |
| 4 | 47.36 | 18.73 |
| 5 | **47.39** | **18.79** |

Table 13: Analysis on the loss weight $\lambda$ deployed on $\mathcal{L}_{occ}^{aux}$.

| $\lambda$ | IoU | mIoU |
|---|---|---|
| 0.1 | 47.15 | 18.53 |
| 0.2 | 47.36 | **18.73** |
| 0.4 | **47.43** | 18.69 |
| 0.8 | 47.28 | 18.59 |
| 1.0 | 47.39 | 18.42 |

Table 14: Analysis on the loss weight $\lambda_{reg}$ deployed on $\mathcal{L}_{Reg}$.

| $\lambda_{reg}$ | IoU | mIoU |
|---|---|---|
| 0.5 | 47.19 | 18.46 |
| 1.0 | 47.36 | **18.73** |
| 1.5 | 47.48 | 18.68 |
| 2.0 | 47.55 | 18.59 |
| 2.5 | **47.68** | 18.55 |

## A.5 Sensitivity Analysis

In this section, we further give a detailed analysis of the hyperparameters used in VoxDet. The experiments are conducted with the same random seed to ensure fair comparison. The results are reported on the SemanticKITTI validation set as the unified SSC practice.

**Number of Instance-level Aggregation Layers.** Tab. 12 reports the performance changes as we vary the number of aggregation layers $N$. Performance improves steadily with increasing $N$, confirming the benefit of more effective aggregation. Although extending to five layers (one more than our default $N = 4$) yields a slight additional gain, the marginal improvement will lead to extra computational overhead. We therefore adopt $N = 4$ as our default setting.

**Weight of Auxiliary Loss.** Tab. 13 presents the sensitivity of our model to the auxiliary segmentation loss weight $\lambda$ in $\mathcal{L}_{occ}^{aux}$, which governs the strength of voxel-level supervision. Overall, mIoU remains relatively stable across a wide range of $\lambda$ values: as we increase $\lambda$, performance steadily improves, peaking at $\lambda = 0.2$, and then gradually declines for larger weights. This trend indicates that a moderate auxiliary loss provides beneficial guidance for voxel-wise feature learning, while an overly large weight interferes with the subsequent instance-centric optimization. Consequently, we adopt $\lambda = 0.2$ as our default setting, as it yields the best mIoU.

**Loss weight of Regression.** In Tab. 14, we add a loss weight on the regression $\lambda_{reg}$ and explore the effect of varying the loss weight. A moderate increase in $\lambda_{reg}$ yields further IoU gains, likely due to improved fitting of instance contours. However, excessively large weights cause a slight mIoU drop, probably from conflicts among the combined loss terms. We therefore set $\lambda_{reg} = 1.0$ to balance the effect and eliminate the term in the main paper for convenience.

## A.6 Model Efficiency

Tab 15 presents a comprehensive comparison between our VoxDet and current state-of-the-art SSC methods on the SemanticKITTI test set, evaluating model size, inference speed, and SSC performance. VoxDet requires the fewest learnable parameters, achieves the fastest inference time, and attains the highest per-class IoU as well as the best overall mIoU. Compared to its strongest competitor [2], our approach slashes the parameter count, accelerates inference to real-time performance, and delivers a notable gain in mIoU. These results underscore the strength of our effective SSC formulation, giving a lightweight yet powerful framework that simultaneously advances speed, efficiency, and accuracy.

Table 15: Comparison of model efficiency and accuracy with SoTA on SemanticKITTI test set.

| Venue
Method | CVPR' 23
TPVFormer [21] | CVPR' 23
VoxFormer [34] | CVPR' 24
Symphonize [22] | ICCV'23
OccFormer [82] | IJCAI' 24
StereoScene [25] | NeurIPS' 24
CGFormer [79] | CVPR' 25
ScanSSC [2] | VoxDet |
|---|---|---|---|---|---|---|---|---|
| Param. (M) ↓ | 107 | 59 | 59 | 214 | 117 | 122 | 145 | **53** |
| Inf. Time (ms) ↓ | 207 | 204 | 216 | 199 | 258 | 205 | 261 | **159** |
| IoU ↑ | 34.25 | 42.95 | 42.19 | 34.53 | 43.34 | 44.41 | 44.54 | **47.27** |
| mIoU ↑ | 11.26 | 12.20 | 15.04 | 12.20 | 15.36 | 16.63 | 17.40 | **18.47** |

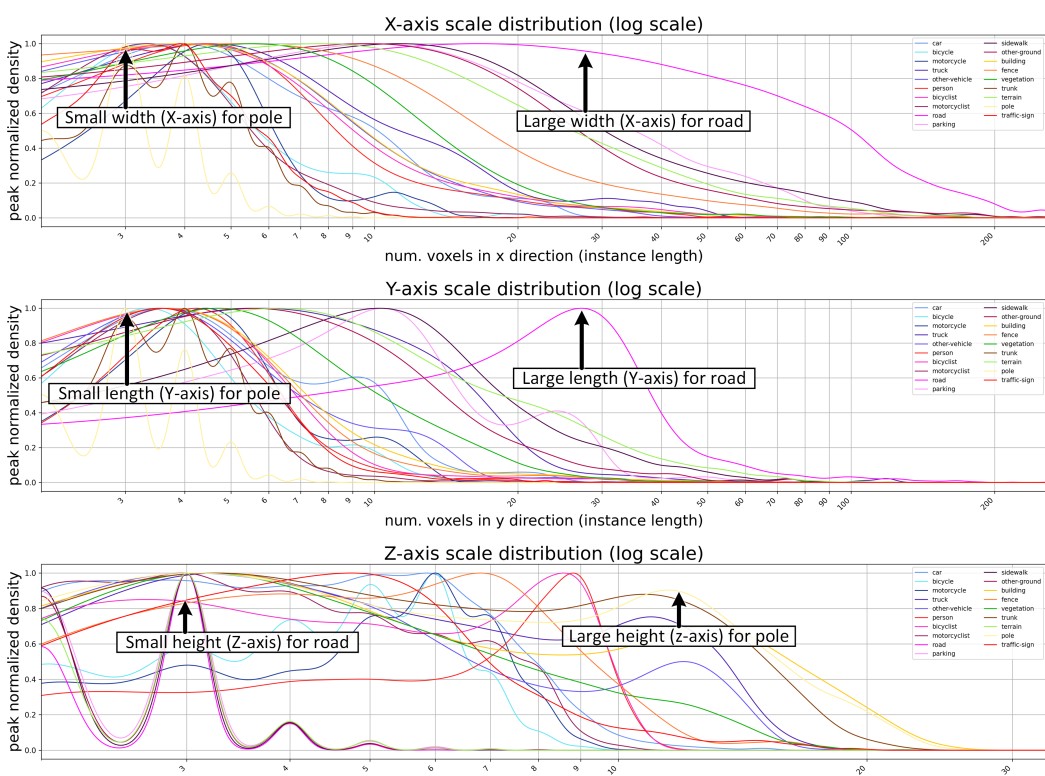

Figure 9: Instance-level scale distribution in $X, Y, Z$ axis given by our voxel-to-instance trick. We randomly sample voxels in all classes and calculate the instance scale ($l$) with coupled offset terms (positive and negative directions), e.g., $l^x = \delta^{x^+} + \delta^{x^-}$ in $X$-axis. The horizontal axis represents the scale (the number of voxels) on a log scale. The scale of the whole scene in $X, Y, Z$ is $256, 256, 32$ respectively. The vertical represents the number of samples, which is peak normalized to $[0, 1]$ by dividing the maximum number of each class for a better view. Zoom in for a better view.

## B  Voxel-to-Instance (VoxNT) Trick

### B.1  VoxNT Trick Can Freely Understand Instance Scales

In Fig. 9, we conduct a statistical analysis on the instance scale along different axes based on the proposed Voxel-to-Instance (VoxNT) Trick. To better visualize the scale patterns, we visualize the distribution for different classes (in different colors), highlighting their specific scale patterns. The resolution of the whole scene is $256 \times 256 \times 32$. To better explore the instance-level geometries, we sum up the positive and negative directions of the 4D offset field $\Delta$ to represent the scale information in different axes. This process is denoted as follows,

$$
\begin{aligned}
l^x &= \delta^{x^+} + \delta^{x^-}; \\
l^y &= \delta^{y^+} + \delta^{y^-}; \\
l^z &= \delta^{z^+} + \delta^{z^-}.
\end{aligned}
\tag{9}
$$

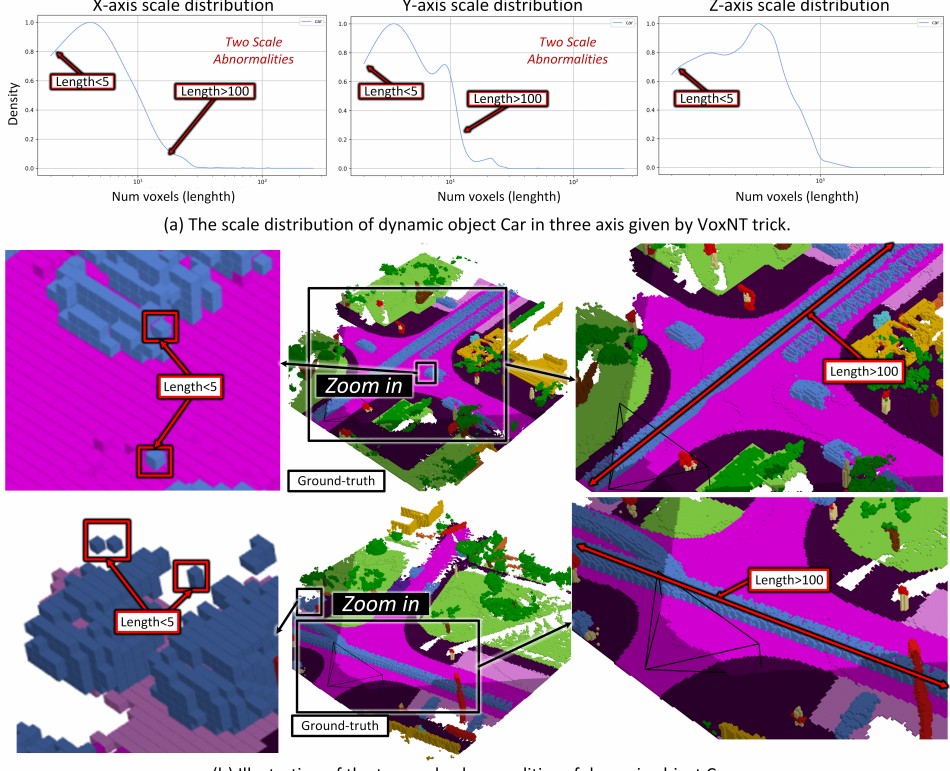

(a) The scale distribution of dynamic object Car in three axis given by VoxNT trick.

(b) Illustration of the two scale abnormalities of dynamic object Car.

Figure 10: More observations from VoxNT. (a) Scale distribution (same as App. B) of the car along the $X, Y, Z$ axis. **We can see two types of abnormality: minimal and large lengths.** (b) Visualizing the abnormality with ground-truth. The left shows the isolated voxels leading to the minimal-scale abnormality. The right shows the long-length abnormality caused by the traces of moving objects.

Here $\{\delta^{x^+}, \delta^{x^-}, \delta^{y^+}, \delta^{x^-}, \delta^{z^+}, \delta^{z^-}\}$ indicates the offset in the six directions. Surprisingly, we find that $\Delta$ is able to freely give sufficient instance-level cues beyond the spatially agnostic semantic labels: the essential scale information. Besides, the geometric scale distribution of different classes also demonstrates different unique patterns, greatly aligning with the real-world environment. The following are intuitive observations.

**(1) For tall instances**, such as *pole* in the light yellow color, the scales tend to give small values in $X$ and $Y$ axes (usually lower than six voxel width/length), and provide a large scale on the $Z$ axis (e.g., larger than 10-voxel height). This is aligned with the comment scene as the width and length of these things are small, while the height is significant.

**(2) For the flat items**, such as *road* in the pink color, we can see a significant value in $X$ and $Y$ axes (usually larger than 30 voxel width/length), while giving a small scale on the $Z$ axis (smaller than five voxels), which also aligns with our intuition.

**(3) For some large objects**, such as *vegetation* in the green color, we can see that the scale in all three axes can be relatively significant, aligning with the real-world environment.

Hence, these observations further highlight the value of the proposed VoxNT trick, which can convey extensive valuable information in geometry at the instance level. Note that this information is not available in the original voxel-level class labels.

## B.2    VoxNT Trick Can Freely Identify Wrong Labels

**Observe the Scale Abnormality of Dynamic Objects with VoxNT Trick.** We first study the *car* class[2], the most prevalent and dynamic object category in autonomous driving. Fig. 10 (a)

---

[2]Cars serve as the key target and dominate the evaluation of detection benchmarks [9, 62]

demonstrates the distribution of the instance scale along different axes, generated by our VoxNT trick. Given the whole scale of $256 \times 256 \times 32$ in the scene, we find that the scale of the car is problematic: In the $X$ and $Y$ axis, there are lots of samples with tiny scale with **length less than 5** and huge scale with **larger than 100**. In the $Z$ axis, many samples have **length less than 5**. Note that the typical scale of a car is less than $30 \times 20 \times 10$, indicating that the ground-truth has significant abnormality.

**Delve into the Abnormality with Label Visualization.** To study the essential reason, we visualize the ground-truth voxels in Fig. 10 (b). We can ground the observed issues mentioned above according to the visualized ground truth, which is voxelized from point clouds. In the left sub-figure, we find that the reason for the samples with abnormally small length is **isolated voxels**. In the right sub-figure, we can see that the failed filter object dynamics lead to abnormally large lengths.

### B.3 VoxNT Trick Can Freely Eliminate the Influence of Wrong Labels in Training

We can see that the scale information in our 4D offset field has a valuable property for solving this issue, which, for the first time, makes the abnormality identification realistic. In this section, we discuss some tricky uses of the proposed VoxNT trick to inspire the follow-up works. Note that these operations improve a more reasonable prediction but cannot improve the mIoU evaluation, because the ground-truth is noisy (see App. B.4). In brief, the key idea is to use scale information to identify the voxel with an offset that is too large or too small. Specifically, to identify the impact of the isolated voxels, we can use a binary voxel-level mask $\mathbf{M}_{i,j,k}^{\min} \in \{0,1\}^{X \times Y \times Z}$ to **identify the voxels that have abnormally small offsets in all axes**:

$$\mathbf{M}_{i,j,k}^{\min} = \begin{cases} 1.0, & (l_{i,j,k}^x < K_{\min}^x) \cap (l_{i,j,k}^y < K_{\min}^y) \cap (l_{i,j,k}^y < K_{\min}^z); \\ 0.0, & \text{otherwise,} \end{cases} \tag{10}$$

where $K_{\min} \in \mathbb{N}^3$ is a scale threshold for the three axes for measuring the removed minimal scale and can be empirically set to 3. Similarly, to filter out the huge-scale samples, e.g., the car in Fig. 10, a similar mask can be deployed $\mathbf{M}_{i,j,k}^{\max} \in \{0,1\}^{X \times Y \times Z}$ to **identify the voxels with abnormally large offsets appearing in one of the axes**. This can be written as follows,

$$\mathbf{M}_{i,j,k}^{\max} = \begin{cases} 1.0, & (l_{i,j,k}^x \geq K_{\max}^x) \cup (l_{i,j,k}^y \geq K_{\max}^y) \cup (l_{i,j,k}^y \geq K_{\max}^z) \\ 0.0, & \text{otherwise,} \end{cases} \tag{11}$$

where $K_{\max}$ is the threshold filtering out the large lengths. Another class-based mask can also be adopted $\mathbf{M}_{i,j,k}^{\max}$ with class labels $Y_{i,j,k}$ and desired class $K_c$ for the wrong label filtering, such as car.

By using these two types of masks, we can clearly identify and localize those wrong voxels with the specific positions $(i, j, k)$ in the volumes. To remove their influence on training, it is possible to directly refine the ground-truth labels by ignoring these voxels with a straightforward yet effective label transformation. In practice, it can be implemented as the following equations,

$$Y_{i,j,k}^{\text{refined}} = \begin{cases} 255, & (\mathbf{M}_{i,j,k}^{\max} = 1) \cup (\mathbf{M}_{i,j,k}^{\min} = 1) \cap (Y_{i,j,k} = K_{car}); \\ K_{car}, & (\mathbf{M}_{i,j,k}^{\max} = 0) \cap (\mathbf{M}_{i,j,k}^{\min} = 0) \cap (Y_{i,j,k} = K_{car}); \\ Y_{i,j,k}, & \text{otherwise.} \end{cases} \tag{12}$$

Here, the key idea is to set the label as ignored (255) if the scale of the car voxel is too large or too small. This can effectively remove lots of wrong predictions in the dynamic car.

### B.4 Rethink the Evaluation on Dynamic Objects

Based on the observation about wrong labels, we further delve into the SSC evaluation on the dynamic object *car*. In Fig. 11, we visualize the ground-truth of the sample from SemanticKITTI [3] validation (Left), the corresponding prediction from the state-of-the-art method [79] (Middle), and the prediction from our method (Right). We can see that previous works are prone to overfitting these wrong labels while generating higher results on the IoU metric for the car. Differently, our method gives a more reasonable prediction. However, this advantage cannot be demonstrated by using the conventional IoU metric due to the wrong labels. We hope this observation can inspire the following works to notice and address this issue, thereby pushing forward the community.

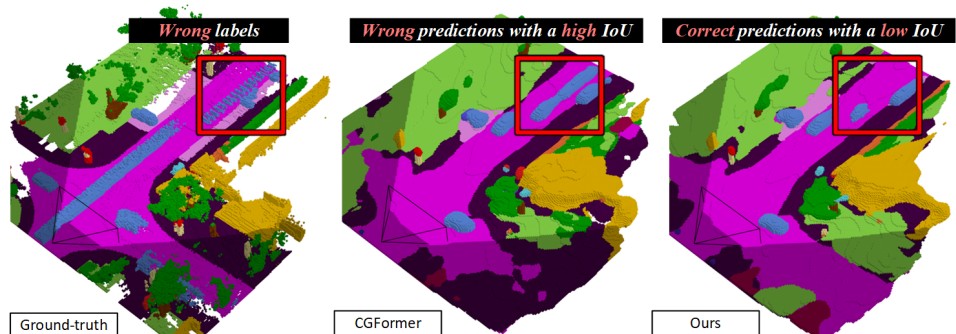

Figure 11: Illustration of the problems in the existing evaluation metrics. **Left:** The ground-truth wrongly labels the car with a sequential afterimage. **Middle:** Previous works wrongly fit these errors by predicting a long car. **Right:** Differently, our VoxDet can give more reasonable predictions without fitting the error, but the car IoU in this sample is worse due to the wrong labels.

## C  Experimental Settings

### C.1  Datasets and Metrics

**Benchmark Setting.** Following the unified practice in SSC, we evaluate our approach on two benchmarks: SemanticKITTI [4] and SSCBench-KITTI-360 [33], which are derived from the original KITTI Odometry [16] and KITTI-360 [37] datasets, respectively. In all experiments, we follow the unified setup [34, 79, 22], restricting to the frustum of size $51.2\,\mathrm{m}$ in the forward direction, $\pm25.6\,\mathrm{m}$ laterally, and $6.4\,\mathrm{m}$ vertically above the sensor. This volume is voxelized into a grid of size $256 \times 256 \times 32$, with each voxel measuring $0.2\,\mathrm{m}$ on a side. The dataset details are as follows.

(1) SemanticKITTI consists of 22 sequences (00–21) of LiDAR scans and synchronized stereo images. We adopt the standard split of 10 sequences for training (00–07, 09–10), one sequence for validation (08), and 11 sequences for online evaluation on the hidden test server (11–21). Input images are provided at $1226 \times 370$ resolution, and ground-truth occupancy grids are annotated with 20 labels (19 semantic classes + 1 empty class).

(2) SSCBench-KITTI-360 is a recently released extension that re-labels a subset of the KITTI-360 sequences for SSC. It comprises 9 sequences in total, of which 7 are used for training, 1 for validation, and 1 held out for final testing. RGB images are captured at $1408 \times 376$ resolution, and each voxel is annotated with one of 19 labels (18 semantic categories + 1 free-space class).

**Evaluation Metrics.** We assess performance along two complementary axes, i.e., geometry completion and semantic completion, using the Intersection over Union (IoU) and mean Intersection over Union (mIoU) metrics, for occupied voxel grids and voxel-wise semantic predictions. All reported results follow the evaluation protocols in SSC, including the online test-server evaluations for SemanticKITTI [4], ensuring a fair comparison with prior methods [34, 21, 79].

### C.2  Implementation Details

**Backbone Network.** Following the main stream of SSC works [34, 83], we use ResNet-50 [20] as the 2D image feature extractor. We follow the unified SSC settings in the recent 2-year publications for the depth estimation and use MobileStereoNet [52] and Adabins [5] as the depth estimators. Note that the compared state-of-the-art methods use the same depth images for the fair comparison.

**View Transformation.** In the main paper, the 2D-to-3D view transformation follows [79, 2]. Specifically, given the input image $\mathbf{I}$, we first extract 2D image feature $\mathbf{F}^{2D}$ and depth map $\mathbf{Z}$. Then, we adopt a depth refinement module that takes the 2D feature $\mathbf{F}^{2D}$ and depth $\mathbf{Z}$ as input. It first estimates the depth distribution via LSS [46], and then generates voxel queries $\mathbf{V_Q} \in \mathbb{R}^{X \times Y \times Z \times C}$. Here, $(X, Y, Z)$ is spatial resolution $128 \times 128 \times 16$, and $C = 128$ is the channel.

Based on this, the pixel $(u, v)$ can be transformed to the 3D point $(x, y, z)$ using the camera intrinsic matrix $(\in \mathbb{R}^{4 \times 4})$ and extrinsic matrix $(\in \mathbb{R}^{4 \times 4})$, termed the query proposals $\mathbf{Q}$ in the projected voxel space. Finally, the 3D deformable cross-attention is deployed to query the information from 2D

image to the 3D voxel space. The number of deformable cross-attention layers is 3, and the number of sampling points around each reference point is set to 8. This can generate the 3D feature volume with dimensions of $128 \times 128 \times 16$ and 128 channels, which is the $\mathbf{V}$ in the main paper. For fair comparison, these operations in view transformation are the same as previous works [79, 2, 60]. Kindly refer to [79] for more details.

**VoxDet.** The number of instance-driven aggregation layers $N$ is set to 4. The loss weight terms of $\lambda$ and $\beta$ are empirically set to 1.0 and 0.2, respectively. For the task-shared voxel encoder in our spatially-decoupled voxel encoder, we directly use the encoder part of the conventional 3D UNet in other SSC works [79]. The regression branch is simply deployed as $\texttt{Conv} \rightarrow \texttt{GroupNorm} \rightarrow \texttt{ReLU} \rightarrow \texttt{Conv}$, with the last convolution outputting 6 channels. Code and models will be released.

**Model Training.** We train our VoxDet with a batch size of 4 using AdamW [42] optimizer. Following [79], the cosine annealing schedule is adopted, with the first $5\%$ iterations of warm-up, maximum learning rate of $3 \times 10^{-4}$, weight decay of $0.01$ and $\beta_1 = 0.9$, $\beta_2 = 0.99$, The experiments are conducted on 2 NVIDIA A100 GPUs (40G) with two samples for each GPU. The final prediction has dimensions of $128 \times 128 \times 16$, which is upsampled to $256 \times 256 \times 32$ through trilinear interpolation to align with the ground truth. We use the VoxNT trick to remove the wrong labels as Eq. 12, with the scale threshold set to 30 for each axis. The efficiency experiments are conducted on a NVIDIA 4090 (commercial GPU), considering the more practical deployment property. The training requires around 19 GB of memory per sample, with about 9.0 training hours for SemanticKITTI and 18 hours for SSCBench-KITTI360, which is very friendly for the research groups with commercial GPUs.

**Implementing VoxDet with LiDAR Input.** VoxDet is designed on the lifted 3D volumes, which can be effortlessly transferred to a LiDAR-based pipeline. To further analyze this flexibility, we deploy a LiDAR-based VoxDet, denoted as **VoxDet-L**. We achieve this by replacing the 2D-to-3D lifting with a simple point cloud encoder. This can directly generate the 3D feature volume $\mathbf{V}$ with 3D ResNet-50, using LiDAR point cloud as input. The 2D image encoder, depth estimator, and view transformation are removed. *All the implementations only use a single model without multi-frame distillation and only a single frame input without temporal information,* which will be open-sourced.

### C.3 Algorithmic details of the VoxNT Trick

We present the detailed process of our VoxNT Trick in Algorithm 1 with PyTorch-style pseudo code. By calling the function $\text{compute\_all\_direction\_distances}()$, this algorithm aims to generate the ground-truth of 4D offset field $\hat{\Delta}$ only using the per-voxel class labels $Y \in \mathbb{N}^{X \times Y \times Z}$ (gt\_occ). In $Y$, each item satisfies $0 < Y_{i,j,k} < K$ with $K$ representing the number of classes.

In brief, for every scanning direction $\mathbf{d} \in \{x^+, x^-, y^+, y^-, z^+, z^-\}$, we begin by initializing a zero-valued matrix $R \in \mathbb{N}^{X \times Y \times Z}$. Then, as we iteratively traverse the voxels along the chosen direction $d$, we compare consecutive voxels. Given the two adjacent voxels $\mathbf{V}_i$ and $\mathbf{V}_{i+1}$, suppose the class label of them matches with each other: $Y_i = Y_{i+1}$ considering the scanning direction, the corresponding entry in $R$ is incremented by one; otherwise, the accumulation halts once a class change is detected, revealing approaching the instance boundary.

This trick efficiently captures the spatial continuity of object instances and provides reliable ground truth offsets for regression. Considering the symmetrical property between the two scanning directions (forward and backward), we deploy a flip operation along the selected axis for efficient implementation. Based on this, we can generate the 4D tensor saving these relative distances in 6 directions, where each dimension is then normalized into $[0, 1]$ divided by the volume size $X, Y, Z$ accordingly. Thus, we can obtain a normalized offset field $\hat{\Delta}$ ground-truth from semantic labels.

## D Discussion

### D.1 Difference with Related Works

**Occupancy prediction assisted with 3D object detection.** Some existing works have attempted to assist the voxel prediction with 3D object detection [45, 75]. Although they consider the instance-level representation, our VoxDet is essentially different from these works in the following aspects.

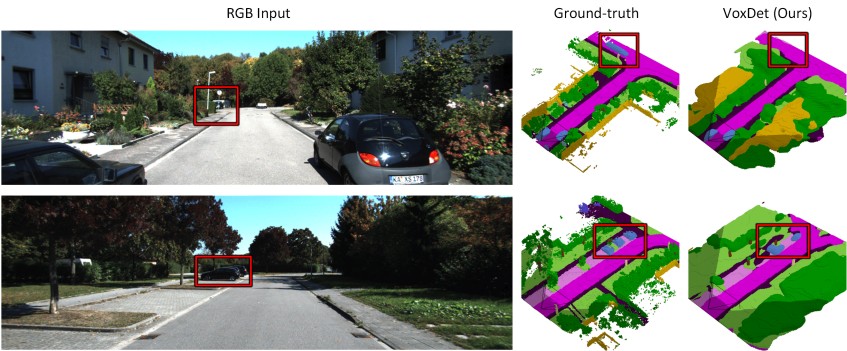

Figure 12: Failure cases of the our VoxDet. It is difficult for our method to detect objects satisfactorily and correctly at extremely far distances due to limited semantic cues on the RGB input images.

**(1) Extra-label Free.** Different from these works that require extra instance-level bounding box labels, VoxDet does not need any handcrafted bounding box labels for the object instances thanks to the proposed VoxNT trick, which significantly reduces the labeling labor with more practical usage.

**(2) Extension to Image and LiDAR-base Pipelines.** Only using SSC labels, VoxDet can be extended to both camera-based and LiDAR-based settings in a unified pipeline, achieving state-of-the-art results on both benchmarks. This differs from most existing works, which are tailored only for a single type of input modality. Hence, our method has more substantial flexibility and transferability.

**(3) Flexible Definition of Object Instances.** These works [45, 75] highly rely on the specific definition of instances in the detection datasets, which usually only considers some primary objects like cars and humans. However, extending to a more profound class set, such as buildings, is challenging and almost impossible, making this instance-level perception un-extensible. Differently, our VoxNT can handle the instances in arbitrary classes, providing an extremely flexible definition for object instances. For example, we can discover the length, width, and height of buildings, vegetation, etc (see Fig. 9), which is highly valuable for autonomous navigation.

**(4) Technical Difference.** Existing works [45, 75] introduce a separate 3D-object-detection branch to support occupancy prediction. In contrast, we consolidate both tasks into a single, detection-driven formulation with an ultra-lightweight design, which can avoid the potential conflict between occupancy prediction and 3D object detection. Our feature decoupled designs also avoided the task-misalignment issue, which is ignored in existing works. The proposed dense-regression design, i.e., each voxel regresses the distance to the instance boundary, is a different paradigm for instance-level perception, which is implemented purely with convolutional layers, eliminating the complexity and computational overhead of object-query–based attention.

### D.2 Broader Impacts

**Impacts to the Broader Occupancy Community.** Our VoxNT and VoxDet can be directly used in any voxel-included settings, such as multi-view occupancy prediction, as it is based on a similar voxel representation. By freely transferring semantic voxel labels into instance-level offset labels, this work may also inspire the occupancy community to reconsider the instance-level perception (e.g., instance and panoptic segmentation) achieved using only semantic labels, potentially contributing to training scale-up with practical label usage.

**Impacts to the 3D Point Cloud Community.** As voxelization is a key procedure in point-cloud understanding, our method, fully based on voxel representation, has great potential for the point cloud community. We achieve state-of-the-art results on the LiDAR-based SemanticKITTI benchmark, which justifies our potential impact on the point cloud community.

**Impacts to the 3D Object Detection Community.** Our VoxNT bridges the gap between dense occupancy and sparse 3D object detection. It converts semantically rich voxel-wise labels (e.g., car, building, vegetation) into instance-level offset labels that capture precise object extents and class-aware geometric priors. By integrating these instance labels into the occupancy, 3D detectors can revisit their image-driven origins, i.e., detecting objects directly from 2D inputs, while evolving toward a voxel-based instance detection paradigm that more faithfully reflects the physical 3D world.

### D.3 Limitations and Future Work

**Generate Sparse 3D Bounding Box Visualization.** The current VoxDet uses instance-level supervision to guide the dense SSC tasks, which can not directly generate spare 3D bounding boxes. The reason is that in dense detection [56, 39, 23], the Non-Maximum Suppression (NMS) is required to remove low-quality boxes for final visualizations. However, there is a gap between the 2D pixels and 3D voxels. To solve this, we will develop a 3D NMS algorithm tailored for voxels, bridging the gap between the voxel-level occupancy prediction and instance-level 3D object detection.

**Lack the Usage of Extra Temporal Information.** The current VoxDet is based on single-frame input. While state-of-the-art, the performance can be further improved with extra temporal information [24, 34], which can correct the cross-frame inconsistency. This will be our future work.

**Implementation on Multi-View Pipelines.** The current VoxDet is evaluated with a single camera input, which may limit its application scope. Our method can be transferred to the multi-view settings effortlessly, as it is deployed on the 3D feature volume, which is also the representation for multi-view occupancy prediction. This will be our future work.

**Rely on the Depth Prediction Accuracy.** Although VoxDet achieves state-of-the-art performance, we empirically find that it will also suffer from some false-negative predictions of the objects at a long distance (see App. E). The sub-optimal depth estimation may be the reason. In the main paper, we also find that replacing the stereo depth with monocular depth leads to some performance drop, highlighting the reliance on depth accuracy. Therefore, improving the depth models in the future may further enhance the performance and serve as our future work.

### D.4 Ethical Claims

Our VoxDet uses only publicly available datasets and produces 3D semantic volumes without retaining any personally identifiable information. We do not foresee any significant negative impacts: the method does not enable individual tracking or intrusive surveillance, poses no additional privacy or safety risks beyond standard camera perception systems, and does not rely on sensitive demographic attributes. By focusing on high-level scene understanding for benign applications such as autonomous navigation, our approach raises no known ethical, fairness, or regulatory concerns.

## E  Additional Qualitative Results

### E.1  Failure Cases

In Fig. 12, we present some failure cases generated by the proposed VoxDet. We observe that our method fails to give correct detection for the object at extremely far distances (top sample), especially when objects of similar color are highly overlapped in the 2D images (bottom sample). This may be caused by the minimal visual cues on the 2D images, limiting the capacity. Some potential solutions may be (1) deploying more powerful visual feature extractors tailored for high-quality dense perception; (2) introducing extra modality, such as LiDAR, to enhance the information in the far distance; (3) developing tailored algorithms refining the features of objects in the far distance.

### E.2  More Comparison with Other Methods

In Fig. 13 and 14, we provide additional qualitative comparisons against the state-of-the-art methods CGFormer [79] and OccFormer [82]. Our approach exhibits visually superior instance-level completeness, greater environmental consistency, and enhanced semantic perception.

**Algorithm 1** PyTorch Style Pseudocode of Voxel-to-Instance (VoxNT) Trick.

```python
def compute_all_direction_distances(gt_occ):
    B, X, Y, Z = gt_occ.shape

    dist_x_pos = run_length_along_dim(gt_occ, 1, "positive")
    dist_x_neg = run_length_along_dim(gt_occ, 1, "negative")
    dist_y_pos = run_length_along_dim(gt_occ, 2, "positive")
    dist_y_neg = run_length_along_dim(gt_occ, 2, "negative")
    dist_z_pos = run_length_along_dim(gt_occ, 3, "positive")
    dist_z_neg = run_length_along_dim(gt_occ, 3, "negative")

    # Stack into shape (B, 6, X, Y, Z)
    distances = torch.stack([
        dist_x_pos, dist_x_neg,
        dist_y_pos, dist_y_neg,
        dist_z_pos, dist_z_neg
    ], dim=1)
    return distances

def run_length_along_dim(t, dim, direction):
    if direction == "positive":
        return run_length_positive(t, dim)
    else:
        # flip, compute positive, then flip back
        tf = torch.flip(t, dims=(dim,))
        of = run_length_positive(tf, dim)
        return torch.flip(of, dims=(dim,))

def run_length_positive(t, dim):
    shape = t.shape
    L = shape[dim]
    out = torch.empty_like(t, dtype=torch.int32)

    # Initialize the last slice along dim to 1
    idx_last = [slice(None)] * len(shape)
    idx_last[dim] = - 1
    out[tuple(idx_last)] = torch.tensor(1, dtype=torch.int32, device=t.device)

    for i in range(L - 2, - 1, - 1):
        idx = [slice(None)] * len(shape); idx[dim] = i
        idx_next = [slice(None)] * len(shape); idx_next[dim] = i + 1

        current = t[tuple(idx)]
        nxt = t[tuple(idx_next)]

        # Check whether the next voxel is in the same class
        same = (current == nxt)
        out_next = out[tuple(idx_next)]

        out_val = torch.where(
            same,
            out_next + 1,
            torch.tensor(1, dtype=torch.int32, device=t.device)
        )
        out[tuple(idx)] = out_val

    return out
```

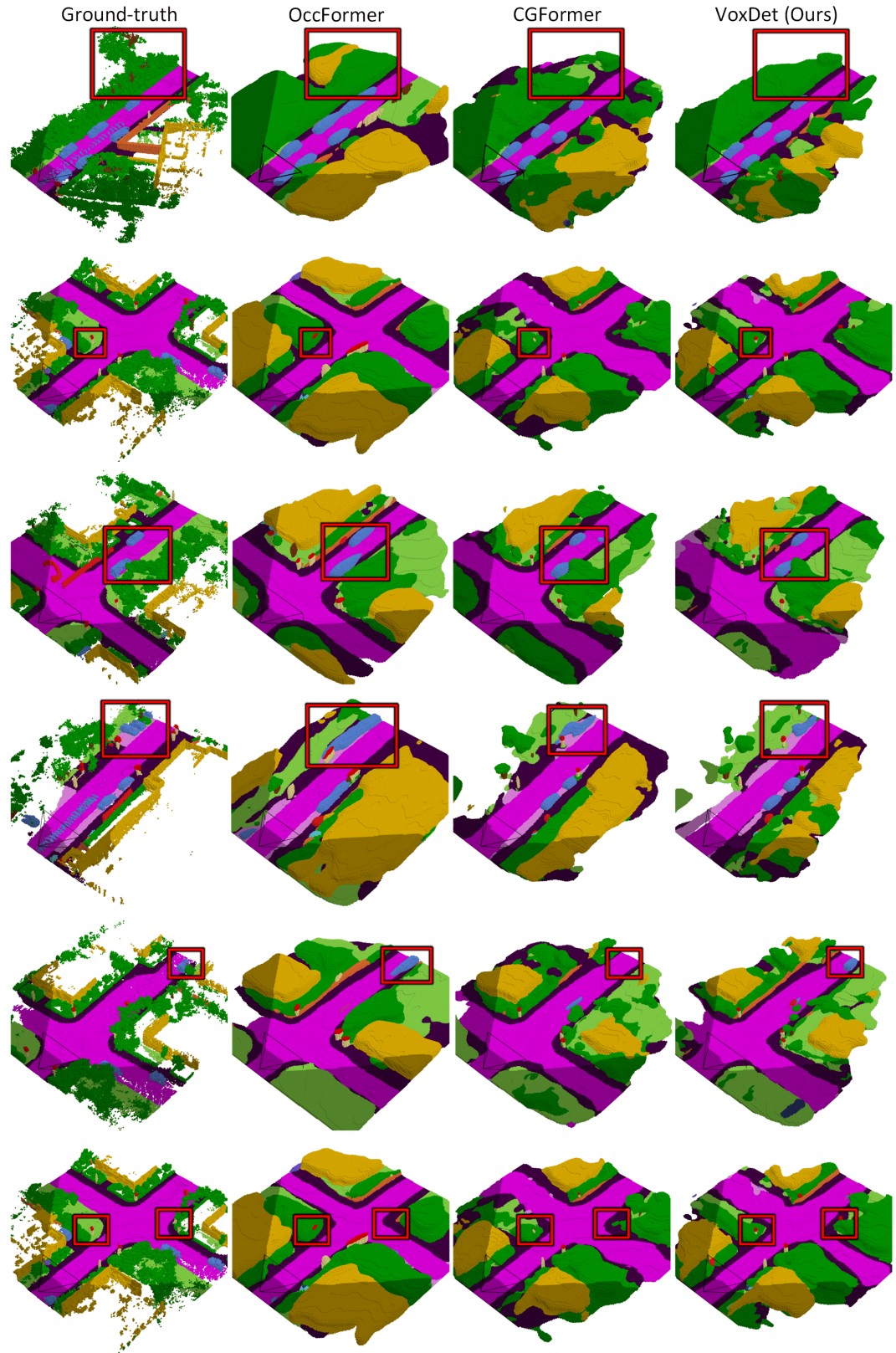

Figure 13: More qualitative comparisons on SemanticKITTI validation set.

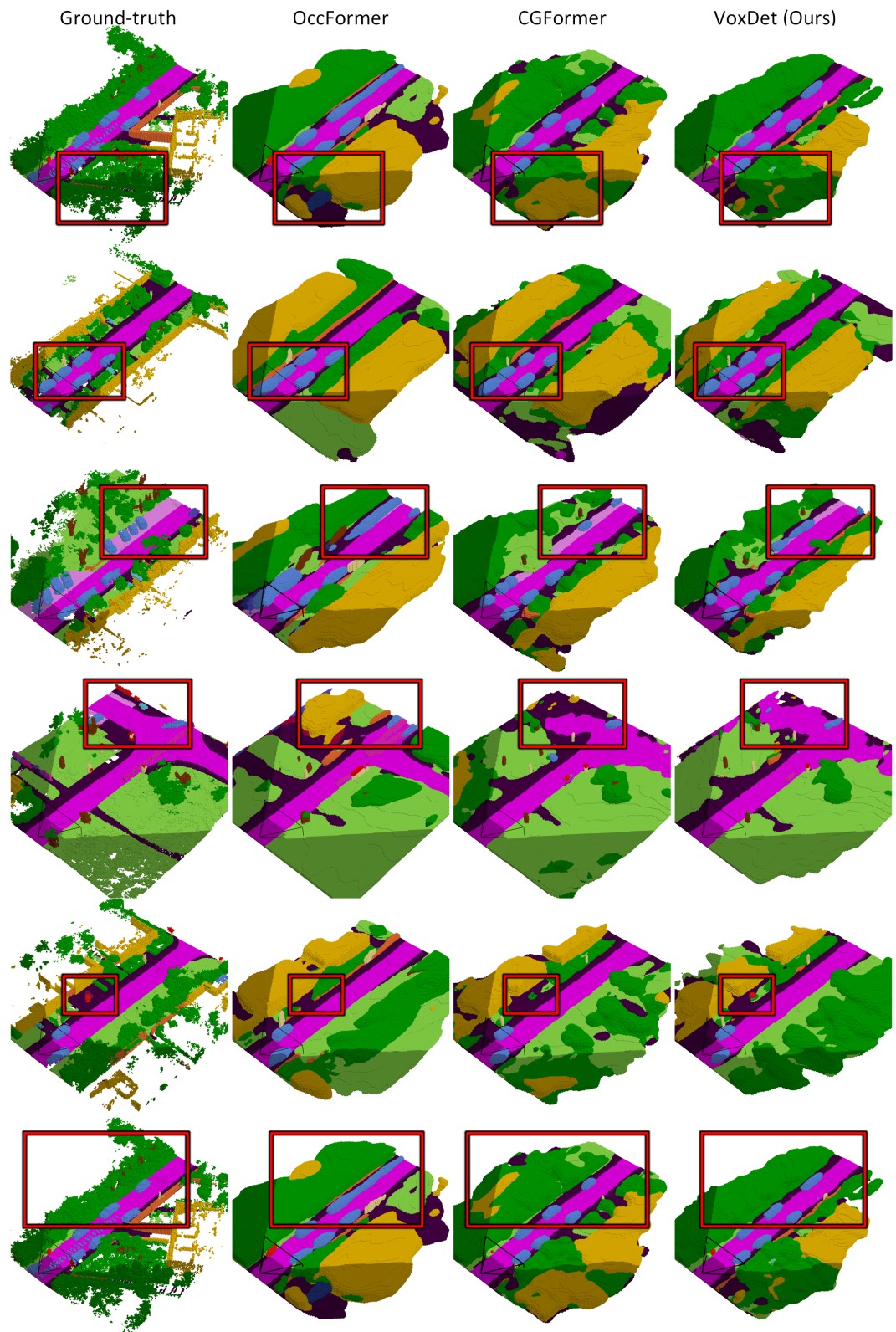

Figure 14: More qualitative comparisons on SemanticKITTI validation set.

