# OpenReview forum: "VoxDet: Rethinking 3D Semantic Scene Completion as Dense Object Detection"
_NeurIPS.cc/2025/Conference — NeurIPS 2025 spotlight_

### Official Review · Reviewer_8beq · 2025-06-13

**Clarity:** 3
**Significance:** 4
**Originality:** 4
**Rating:** 6
**Confidence:** 5

**Summary:**

This paper aims to tackle semantic scene completion (SSC), which reconstructs 3D semantic occupancy from camera input. Starting from an interesting observation that the object instance can be regressed in the voxel space, the authors first propose a simple yet effective VoxNT trick to freely convert the voxel-level class labels to the instance-level offsets. Leveraging these free offset labels, they further propose VoxDet, an instance-centric framework that reformulates SSC as a dense object detection task with two subtasks: semantic prediction and offset regression. VoxDet first employs a spatially decoupled voxel encoder to learn task-specific volumetric representations via distinct spatial deformations. Then it introduces a task-decoupled dense predictor to regress instance borders and aggregate instance semantics. Extensive experiments on both camera and LiDAR benchmarks show that VoxDet achieves new state-of-the-art results jointly, which also gives the highest 63.0 IoU on the SemanticKITTI hidden test set (online leaderboard). Overall, this work is both exciting and inspiring for the community.

**Questions:**

Overall, this is an interesting paper. All questions are listed above, I would appreciate if those issues can be addressed.

**Ethical Concerns:**

["NO or VERY MINOR ethics concerns only"]

**Final Justification:**

Thank the authors for their response, which addressed my questions. After carefully reading the other reviews and the rebuttal, I find that the authors have not only provided strong clarifications but also highlighted additional advantages of their method, including (a) strong performance and generalization ability on other datasets, (b) superior problem formulation and novelty compared to previous works, (c) robustness to different voxel resolutions without ethical concerns, and (d) easy-to-reproduce implementation details, with the core source code available.

Overall, I believe this work is insightful and high-quality, which can make significant contributions to the occupancy community.

**Limitations:**

Yes.

**Quality:**

3

**Strengths And Weaknesses:**

Strengths

- The paper is clearly organized and well-motivated with good technical flow.
- Section 3 offers practical, interesting observations that clearly motivate the core method.
- The proposed VoxNT trick cleverly extracts instance-level scale information from voxel-level class labels without extra supervision and training.
- In VoxDet, the reformulation of SSC as dense object detection is both novel and effective, providing new insights and perceptive. The detailed algorithm designs are also tailor-designed and technically sound.
- The approach is computationally efficient and delivers state-of-the-art results on both camera and LiDAR benchmarks, with the highest IoU on the online leaderboard. The multi‐run experiment in appendix demonstrates good robustness.
- A comprehensive experimental suite, such as feature visualizations and quantitative studies, supports the method design and yields additional insights.


After a thorough review of the main paper and appendix, I found no major weaknesses and have only the following minor question:

- In the appendix, the method achieves remarkable performance on LiDAR benchmark, which is suggested to reported in the main paper. Accordingly, given the strong LiDAR-based results, I recommend reframing the paper’s focus from camera-based SSC to the more general SCC task across sensing modalities.
- Can the proposed method be further improved by using both camera and LiDAR inputs together?

---

> ### Author Rebuttal · Authors · 2025-07-30
>
> We sincerely appreciate the careful review of the details in both the manuscript and the Appendix. The comments precisely captured the key aspects we hoped to convey, including the technical insight, methodological novelty, and experimental validation of our work, which encourages us significantly.
>
> ## [Q1] Move LiDAR-based results from the Appendix to the main paper
>
> We fully agree with this constructive suggestion and will make the change in the final version. Bringing the LiDAR results into the main paper will better highlight two core contributions: (1) VoxDet is a truly general method that works across diverse datasets and **can be deployed with either camera or LiDAR input**, which is a unique technical highlight not offered by prior SSC works. (2) VoxDet achieves **state-of-the-art performance on both camera and LiDAR settings**, ranking 1st on the public leaderboard, justifying the strong potential for general usage.
>
> ## [Q2] Can the method be further improved by fusing both LiDAR and camera modalities?
>
> Thank you for your insightful suggestion! Yes, our method **can be effortlessly deployed using camera and LiDAR input together**. We further evaluate our approach on SemanticKITTI by fusing two modalities: camera-based 3D volumes (after 2D-to-3D lifting) and LiDAR-encoded 3D volumes via an element-wise addition. As shown below, this multimodal fusion boosts mIoU from 26.0% to 26.9%. These results demonstrate that incorporating the camera modality, with its rich RGB cues, significantly enhances semantic understanding of VoxDet, further highlighting the adaptability and effectiveness of our method.
>
> | Method           | Reference    | IoU  | mIoU  |
> |:-----------------|:----------------|:----------|:-----------|
> | VPNet           |   NeurIPS 2024      |  60.4   | 25.0    |
> | VoxDet (LiDAR)           |  Ours          |  63.0   |    26.0    |
> | VoxDet (LiDAR-Camera-Fusion)           |   Ours         | **63.2**    | **26.9** |

---

> > ### Comment · Reviewer_8beq · 2025-08-09
> >
> > Good job. I vote for acceptance. I have given the final rating several days before.

---

### Official Review · Reviewer_6r4q · 2025-07-01

**Clarity:** 3
**Significance:** 3
**Originality:** 3
**Rating:** 5
**Confidence:** 4

**Summary:**

This work introduces a semantic scene completion algorithm based on voxel-to-instance class label offset prediction. After 2D to 3D lifting, the framework conducts two separate spatial decoupling modules, where one branch predicts voxel semantic labels while the other aims to capture instance-level information of each voxel. The learned instance-level voxels are integrated into the semantic-level voxels for scene completion. The authors have conducted exhaustive experiments across multiple benchmarks and comparisons with various baselines.

**Questions:**

See weaknesses.

**Ethical Concerns:**

["NO or VERY MINOR ethics concerns only"]

**Final Justification:**

The detailed explanations and response have resolved my concerns, and I will maintain my score.

**Limitations:**

See weaknesses.

**Quality:**

3

**Strengths And Weaknesses:**

**Strength**
- The idea is simple yet effective.
- The experiments are quite strong. The authors have conducted extensive ablation studies and analysis of different designs of different modules.
- The paper is well-written, with proper content for high-level motivations, low-level technical details, and clear figures.

**Weaknesses**
- The algorithm relies on additional instance-level supervision, which is not always available in the training datasets. Is it possible to leverage existing instance segmentation algorithms (SAM) to create such labels? How large a negative effect would it have on the system?
- The authors are expected to discuss some existing instance detection algorithms [1,2] in the related works, as they do complementary research (integrating 3D knowledge for 2D tasks).
- Though this algorithm is designed for outdoor scene completion. I am wondering if it is also applicable to indoor scenarios [3,4]? And if it does not, what would be the key bottleneck / potential ways to address it?

[1] Li, Bowen, et al. "VoxDet: voxel learning for novel instance detection." Advances in Neural Information Processing Systems 36 (2023): 10604-10621.
[2] Shen, Qianqian, et al. "Solving instance detection from an open-world perspective." Proceedings of the Computer Vision and Pattern Recognition Conference. 2025.
[3] Straub, Julian, et al. "The replica dataset: A digital replica of indoor spaces." arXiv preprint arXiv:1906.05797 (2019).
[4] Dai, Angela, et al. "Scannet: Richly-annotated 3d reconstructions of indoor scenes." Proceedings of the IEEE conference on computer vision and pattern recognition. 2017.

---

> ### Author Rebuttal · Authors · 2025-07-30
>
> We sincerely appreciate the constructive comments. We want to summarize and highlight three key clarifications.
> 1. Our method **requires no additional manually annotated instance-level labels** and can be further boosted with 2D foundation models such as SAM.
> 2. We will broaden our related-work section to include a discussion of instance-detection research.
> 3. Our approach offers a general and powerful SSC paradigm that achieves **state-of-the-art results on almost all existing SSC benchmarks (4 datasets)**, across (1) indoor and outdoor scenes and (2) camera and LiDAR-based settings.
>
> We would be grateful if these clarifications prompt a reconsideration of our score, and we remain fully available for any further questions during the author-reviewer discussion period.
>
> ## [W1] Clarify using extra instance labels and using SAM to generate labels
>
> **No additional instance-level labels needed.** We are sorry for the confusion. We'd like to kindly clarify that our VoxNT trick (Fig. 4 in the main paper) automatically generates instance-level pseudo-offset labels from the standard semantic occupancy annotations used in previous SSC works. This process requires **no extra manual instance-level labeling** and thus preserves strict fairness in data usage.
>
> **Use SAM to create pseudo labels.**  We appreciate your insightful suggestion! To validate the potential of external instance-level labels, we conducted an additional experiment using Grounded SAM [#1] to produce 2D instance pseudo labels. Specifically, for each image, we first apply Grounded SAM to generate pseudo instance masks corresponding to the defined classes in the SSC datasets. We then follow previous SSC work [#2] to introduce an auxiliary MaskDINO [#3] head after the ResNet-50 backbone as an additional instance prediction head, supervised by the instance-level loss functions of MaskDINO. It is worth noting that this MaskDINO head is used only during training to enhance instance-aware representation of the backbone, which **does not incur extra parameters or computational overhead at inference time**.
>
> In the table below, we present a comprehensive comparison with existing state-of-the-art methods. After incorporating the labels from Grounded SAM, our method achieves consistent performance gains in both IoU and mIoU, surpassing previous works on all metrics. These results indicate that our approach effectively leverages the external knowledge and labels provided by 2D foundational models, verifying its effectiveness. Therefore, **using SAM to generate labels not only has no negative impact but also further boosts performance**.
>
>
> | Method           | Reference    | IoU ↑ | mIoU  ↑ | Parameters ↓ | Inference Time ↓ |
> |:-----------------|:----------------|:----------|:-----------|:-----------|:-----------|
> | CGFormer           |   NeurIPS 2024        |    45.99|      16.89 | 122 | 205|
> | SGFormer           |   CVPR 2025       |     45.01 |     16.88 |126 | -|
> | ScanSSC           |   CVPR 2025       |     45.95 |      17.12 |145 | 261|
> | DISC           |   ICCV 2025        |     45.93 |      17.05 |-|-|
> | **VoxDet**           |   **Ours**         |    **47.36** | **18.73** | **53** | **159** |
> | **VoxDet w. Grounded SAM**    |    **Ours**  |     **47.92** |      **19.02** | **53** | **159** |
>
> [#1] Grounded SAM: Assembling Open-World Models for Diverse Visual Tasks (ArXiv 2024)
> [#2] Symphonize 3d Semantic Scene Completion with Contextual Instance Queries (CVPR 2024)
> [#3] Mask DINO: Towards A Unified Transformer-based Framework for Object Detection and Segmentation (CVPR 2023)
>
> ## [W2] Discuss some instance detection works
>
> Thank you very much for your suggestion! This makes our manuscript more comprehensive, especially by deepening the review of instance-centric learning, which is the key focus of our work. Li et al. [1] propose a novel 3D-inspired paradigm for geometrically reliable detection in 2D views, enhancing geometric perception through voxel-driven instance reconstruction. Recently, IDOW [2] innovatively adapts foundational knowledge into a 3D-aware instance detection pipeline, significantly improving generalization and robustness in open-world scenarios. Unlike these approaches, which leverage 3D information to augment 2D instance detection, our method discovers object instances directly in dense 3D voxel space, eliminating 2D limitations such as occlusion and perspective distortion. We will revise the final version to review these related works comprehensively.
>
> ## [W3] Explore the adaptability to indoor scenes
>
> Thank you for your constructive suggestions! Our approach represents a generic SSC paradigm adaptable to both indoor and outdoor scenes. To further demonstrate the generalizability of our generic SSC paradigm, we further compare VoxDet on two widely-used indoor datasets: Occ-ScanNet (upper table) and NYU v2 (lower table). On both datasets, VoxDet decisively outperforms all prior SSC methods, underscoring its **ability to adapt to highly diverse voxel settings**. Note that our method even surpasses the co-occurrence works such as DISC and MonoMRN with noticeable gains.
>
> With these new results, VoxDet now sets the state of the art on all four major SSC benchmarks. To the best of our knowledge, **this is the first method to be exhaustively validated across the entire SSC benchmarks**: most prior work reports on at most two of these datasets. This comprehensive superiority highlights our effectiveness and affirms its applicability as a **truly general-purpose solution** for semantic scene completion.
>
>
> | Method           | Reference    | IoU   | mIoU  |
> |:-----------------|:----------------|:----------|:-----------|
> | MonoScene           |   CVPR 2022        |    41.60|     24.62|
> | ISO           |   ECCV 2024       |    42.16 |    28.71 |
> | CGFormer           |   NeurIPS 2024       |  43.02   |29.11     |
> | DISC           |   ICCV 2025              |  43.18   | 28.92      |
> | **VoxDet**           |   **Ours**         |    **46.47** |**30.02**  |
>
>
> | Method           | Reference    | IoU  | mIoU   |
> |:-----------------|:----------------|:----------|:-----------|
> | MonoScene           |   CVPR 2022        |    42.51|    26.94|
> | ISO           |   ECCV 2024       |    47.11 |    31.25 |
> | CGFormer           |   NeurIPS 2024      |  52.72   | 30.97    |
> | MonoMRN           |   ICCV 2025        |  53.16   |   30.73    |
> | **VoxDet**           |   **Ours**         | **55.88**    | **32.15** |

---

> > ### Comment · Reviewer_6r4q · 2025-08-06
> > **Thanks for the response**
> >
> > Thanks for the detailed explainations and response, they have resolved my concerns and I will maintain my score.

---

> > > ### Author Response · Authors · 2025-08-06
> > > **Response to Reviewer 6r4q**
> > >
> > > Dear Reviewer 6r4q,
> > >
> > > Thank you so much for your thoughtful review and constructive feedback, which have been invaluable in helping us improve the quality of our paper. We will carefully incorporate your suggestions into the final version.
> > >
> > > Sincerely,
> > > Authors of Submission #279

---

### Official Review · Reviewer_3a7T · 2025-07-02

**Clarity:** 2
**Significance:** 2
**Originality:** 2
**Rating:** 5
**Confidence:** 4

**Summary:**

The paper proposes a new method for semantic scene completion. It formulates the task as object detection. Given that there are no instance-level labels in the data, the authors propose a heuristic for extracting object instances from semantic labels. It is based on the assumption that individual objects are separable in 3D so there is a change in semantic label at the object boundaries. The method achieves state-of-the-art results on two semantic scene completion benchmarks.

**Questions:**

Do the authors believe that the situations where their assumption is not true are not significant?

Or did I oversee something in the method that makes it robust against wrong supervision due to the proposed heuristic?

Answers to those are extremely important for the method's applicability.

**Ethical Concerns:**

["NO or VERY MINOR ethics concerns only"]

**Final Justification:**

I appreciate the response by the authors and update my rating accordingly. I would kindly ask the authors to discuss the issue in more detail in the main paper and also move the description of the safeguarding mechanism from Appendix B.3 to the main paper.

**Limitations:**

No, see the strengths and weaknesses section.

**Paper Formatting Concerns:**

No concerns.

**Quality:**

2

**Strengths And Weaknesses:**

I acknowledge that the paper achieves state-of-the-art results on two semantic scene completion benchmarks. However, I believe the fundamental assumption behind the method design is wrong and can lead to dangerous consequences on other datasets.

The authors suggest heuristically extracting instance-level labels from semantic labels based on the fact that there is a semantic label change on the object boundaries. This is, however, not always correct. In L.127 they even acknowledge that sometimes "minor ambiguities may arise in cases such as the densely overlapping foliage of adjacent trees". I would argue this is not a minor ambiguity but rather a fundamentally flawed assumption that can be harmful in many real-world situations.

For example if there is a row of cars parked on the side of the road close to each other, they might end up as a single semantic blob when discretized into a voxel grid. The community often works with voxel sizes between 20 and 40 cm so situations like this might not be rare at all. This in turn would lead to the proposed method learning wrong geometric priors. In the context of semantic scene completion this can have dire consequences: the model would to predict a wrong geometry because of the wrong supervision used during training. Automated driving is a safety-critical application so I believe introducing wrong assumptions like this is extremely dangerous.

The fact that this assumption happens to improve the results on some datasets does not justify its use in general. First, because on other datasets those cases where it does not hold true might be statistically more significant. Second, because even a small fraction of safety-critical mistakes introduced by this assumption is enough to refrain from using it.

---

> ### Author Rebuttal · Authors · 2025-07-30
>
> Thank you for raising this concern. We understand the reviewer's point that closely packed objects (e.g., adjacent cars) could, in principle, lead to incorrect instance separation and potentially introduce noisy supervision. We address this in three ways:
>
> 1. **Impact of Incorrect Labels**: In cases where such ambiguities occur, our method will not degrade compared to existing SSC approaches; it simply does not gain the benefit of additional geometric cues. When the instance-level labels are correct (which is the dominant case), our approach yields significant improvements, as evidenced by state-of-the-art results across **four benchmarks** (not just two).
>
> 2. **Empirical Evidence of Robustness**: Despite the theoretical concern, in practice, we observed that even for challenging scenarios (e.g., cars parked with less than 10 cm separation), the heuristic extraction works reliably. This robustness is reflected in consistent performance gains across diverse datasets, which contain many such dense object arrangements.
>
> 3. **Mitigation via Self-Correction (Overlooked Detail)**: Importantly, we already incorporate a **self-correction mechanism** (Appendix B.3) to prevent harmful supervision signals. Specifically, we filter out any pseudo-instance annotations whose size falls outside a reasonable object-size range. As a result, outlier offsets and class labels are automatically ignored during training. This ensures that the model does not learn incorrect geometric priors from erroneous cases.
>
> This mechanism is a unique advantage of our approach: unlike prior SSC methods that implicitly rely on potentially incorrect semantic boundaries without correction, our method actively suppresses invalid labels, thereby reducing risk and enhancing geometric awareness.
>
> In summary, while we acknowledge that the heuristic is not perfect, we have both **empirical evidence** (robust results across multiple datasets) and **built-in safeguards** (size-based filtering) that mitigate the reviewer’s concern. Far from introducing additional risk, our approach makes SSC supervision more informative and inherently safer than existing methods. In the following sections, we provide sufficient evidence and justifications to support the safety, robustness, and generalizability of our method.
>
> ## [Justification 1] Our assumption is free of flaws and better (safer) than prior works
>
> Our VoxNT trick generates pseudo offset labels based on the observation (assumption): **Compared with the 2D image space, object instances are largely free of occlusion in the 3D voxel space; thus, most instances can be discovered to enhance SSC performance, except for tightly overlapping cases** (e.g., intertwined trees). This is a more effective and safer formulation than existing SSC works, which assume **all voxels of the same class as a single class group, totally ignoring per-instance geometry**. Specifically,
>
> 1. For the instances that can be individually discovered, like traffic signs, poles, and most cars (Fig. 6 of the Appendix), **our method can successfully identify their geometric information, which no previous SSC works can achieve.** According to the statistical experiments (Fig. 2 of the Appendix), our algorithm successfully discovers these instances by extracting statistically correct scale offsets.
> 2. For the tightly overlapping cases, like intertwined trees (mentioned in l.127 of the main paper), **our work and all existing SSC methods cannot separate them without real instance labels**.
>
> To better illustrate the safety of our assumption in real-world settings, we list a formal comparison below.
>
>
> | Real-world Scenario                                        | Ground-truth Connectivity        | Prior SSC Work Assumption                                          | Our Assumption                                                    | Safer Choice     |
> |-------------------------------------------------|----------------------------------|----------------------------------------------------------|-----------------------------------------------------------|------------------|
> | 1. Naturally separate (e.g., traffic signs, poles, and trunks) | Separate                         | Single class group                                       | Separate instance groups                                  | Ours             |
> | 2. Naturally merged (e.g., overlapping tree crowns) | Merged                           | Single class group                                       | Fallback to single class group                                  | Equally safe     |
> | 3. Falsely merged (e.g., two cars fused by coarse voxels without air voxels between them) | Merged (e.g., Fig.7 of main paper, sub-optimal label) | Single class group (blindly trusts wrong label)          | Separate instance groups (self-corrected supervision, Appendix B.3) | Ours             |
>
> Hence, compared with previous SSC works, our method demonstrates clear advantages in naturally separate (the dominant case) and falsely merged scenarios, while performing equally well in naturally merged cases. This ensures that our approach does not degrade performance in the complex real-world scenarios, providing a safer and more robust solution.
>
> ## [Justification 2] Our method generalizes well to other datasets
>
> We further conduct experiments on two additional SSC benchmarks: Occ-ScanNet (top table) and NYU v2 (bottom table). In both cases, VoxDet substantially surpasses the current state-of-the-art (including the concurrent work DISC and MonoMRN), showing its adaptability across diverse dataset settings. VoxDet achieves the best performance on all **four major SSC benchmarks** (not just 2), covering both LiDAR and camera settings. To the best of our knowledge, our method has been  **thoroughly evaluated on far more benchmarks and strict settings** than any existing SSC work, fully justifying its general adoption.
>
> | Method           | Reference    | IoU  | mIoU   |
> |:-----------------|:----------------|:----------|:-----------|
> | MonoScene           |   CVPR 2022        |    41.60|     24.62|
> | ISO           |   ECCV 2024       |    42.16 |    28.71 |
> | CGFormer           |    NeurIPS 2024       |  43.02   |29.11     |
> | DISC           |   ICCV 2025              |  43.18   | 28.92      |
> | **VoxDet**           |   **Ours**         |    **46.47** |**30.02**  |
>
>
> | Method           | Reference    | IoU | mIoU  |
> |:-----------------|:----------------|:----------|:-----------|
> | MonoScene           |   CVPR 2022        |    42.51|    26.94|
> | ISO           |   ECCV 2024       |    47.11 |    31.25 |
> | CGFormer           |   NeurIPS 2024      |  52.72   | 30.97    |
> | MonoMRN           |   ICCV 2025        |  53.16   |   30.73    |
> | **VoxDet**           |   **Ours**         | **55.88**    | **32.15** |
>
>
> ## [Justification 3]  Our method is robust to the merged cases
>
> First, to evaluate robustness under the scenarios where more instances are merged, we increase the voxel size from 20 cm to 40 cm by downsampling the voxel resolution (Existing SSC benchmarks use a 20 cm setting). Although this change induces substantially more voxel-level merging (e.g., adjacent vehicles become a single blob), VoxDet outperforms the state-of-the-art method by 3.66 % in IoU and 2.45 % in mIoU. **This justifies that our method is more robust for merging instances, achieving better performance than previous works.**
>
> Second, to further evaluate the correctness of our offset labels generated by the VoxNT trick, we train with the ground-truth instance supervision, denoted as "VoxDet with GT". Compared to training with the real instance-level GT, using our VoxNT-generated pseudo labels to train the model achieves competitive performance, with a performance difference of less than 1%. This demonstrates that our **VoxNT can generate sufficiently accurate instance-level pseudo labels**, even in cases of severe instance merging, fully validating its correctness and robustness.
>
>
> | Method             | Reference       | IoU (40 cm)      | mIoU (40 cm)      | IoU (20 cm)      | mIoU (20 cm)      |
> |:-------------------|:----------------|:-----------------|:------------------|:-----------------|:------------------|
> | CGFormer           | NeurIPS 2024   | 43.01            | 15.72             | 45.99            | 16.89             |
> | VoxDet             | Ours           | 46.67 (+3.66)    | 18.17 (+2.45)     | 47.36 (+1.37)    | 18.73 (+1.84)     |
> | VoxDet with GT     | Ours           | **47.21 (+4.20)**| **18.49 (+2.77)** | **47.64 (+1.65)**| **18.85 (+1.96)** |
>
> ## Response to the rest of the questions
>
> 1. **Do the authors believe that the situations where their assumption is not true are not significant?**
>
>    In fact, our assumption is both rigorous and intuitive, as it explicitly accounts for both separate and merged instances. This makes it applicable to all real-world scenarios. Both cases are significant, and our method consistently performs better than all prior SSC approaches in empirical evaluations.
>
> 2. **Or did I overlook something in the method that makes it robust against wrong supervision due to the proposed heuristic?**
>
>    We apologize for any confusion. We have already designed a self-correction strategy to prevent incorrect supervision when the labels are wrong (see Appendix B.3), which is a unique strength of our method compared to previous SSC works.

---

> > ### Comment · Area_Chair_K2nj · 2025-08-01
> >
> > Dear Reviewer 3a7T,
> >
> > In the first round, you raised an important concern about the potential risk of the method's assumptions.  The authors tried to justify the concerns in the response. Please check the response and provide feedback. Thank you.
> >
> > Best,
> >
> > AC

---

> > > ### Comment · Reviewer_3a7T · 2025-08-05
> > >
> > > I appreciate the response by the authors and update my rating accordingly. I would kindly ask the authors to discuss the issue in more detail in the main paper and also move the description of the safeguarding mechanism from Appendix B.3 to the main paper.

---

> > > > ### Author Response · Authors · 2025-08-05
> > > > **Response to Reviewer 3a7T**
> > > >
> > > > Dear Reviewer 3a7T,
> > > >
> > > > Thank you very much for your review and consideration of the safeguarding aspect, which has helped us further improve the completeness of our work. We greatly appreciate your kind suggestions and will carefully revise the final version to incorporate them.
> > > >
> > > > Sincerely,
> > > > Authors of Submission #279

---

### Official Review · Reviewer_ai3H · 2025-07-05

**Clarity:** 3
**Significance:** 3
**Originality:** 3
**Rating:** 5
**Confidence:** 4

**Summary:**

This paper formulates semantic scene completion (SSC) as a dense object detection task, incorporating offset regression and semantic prediction to address 3D semantic scene completion. The spatially-decoupled voxel encoder and task-decoupled dense predictor are introduced to effectively model task-specific spatial deformation and estimate the 4D offset field. Experimental comparisons demonstrate that the proposed method achieves superior performance compared to other recent SSC methods.

**Questions:**

Tiny weaknesses:

1.	The authors should provide more references to support the assumption “Although prior works, such as Symphonize [16]”
2.	Using (a), (b), (c), and (d) to distinguish the four examples in Figure 2, rather than simply 'left' or 'right,' may be better.
3.	Confusing underlines. For example, “Voxel-level SSC as instance-centric object Detection” in Lines 62-63 of the main manuscript.
4.	The authors should increase the font size of the text in the figures.


The paper reformulates the SSC as a dense object detection task, without sufficient analyses and explanations on the motivations. The missing technical details make the main manuscript hard to follow. Additionally, the authors provide more comparisons and deeper analyses to differentiate their proposed method from previous approaches, which formulate SSC as a dense segmentation task. Therefore, I vote for borderline reject in this review round. I would like to raise the rating if the authors can address my concerns.

**Ethical Concerns:**

["NO or VERY MINOR ethics concerns only"]

**Final Justification:**

The author has addressed my concerns and concerns of other reviewers, and I have updated my score.

**Limitations:**

Yes. The authors have discussed limitations in the supplemental material.

**Paper Formatting Concerns:**

None.

**Quality:**

3

**Strengths And Weaknesses:**

Paper Strengths:

The main insight of this work is the reformulation of SSC. The authors utilize two modules (e.g., spatially-decoupled voxel encoder and task-decoupled dense predictor) to achieve effective semantic scene completion, leading to improved SSC performance.


Paper Weaknesses:

1.	My first concern is the motivation and high-level summary of the reformulation of SSC. The authors reformulate the SSC as a dense object detection, which is different from previous works. However, as demonstrated in Figure 1(b) and technical details (e.g., classification loss) in Section 4.3 Task-decoupled Dense Predictor, the proposed method appears to be similar to previous segmentation-based methods. The authors should provide more analysis to differentiate between these two types of approaches.

2.	My second concern is the technical details. Conv_{(T)}^{(P)} is a 2D convolution module for place (P) and task (T). How many convolutional layers are involved in Conv_{(T)}^{(P)}? Do the different convolution modules share weights or not? It is confusing how the authors utilize these convolution modules to enable effective decoupling. Additionally, the authors utilize FPN to generate V_{cls} and V_{reg}, do the FPNs for these two different tasks have the same network structure and share weights?

---

> ### Author Rebuttal · Authors · 2025-07-30
>
> We sincerely thank you for the careful review, and sorry for the confusion. We would be grateful if these clarifications prompt a reconsideration of our score, and we are actively available for any follow-up concerns during the author-reviewer discussion period.
>
> ## [W1] Clarify the motivation and reformulation
>
> **Motivation.** VoxDet reformulates SSC from per-voxel segmentation (where each voxel only predicts its class based on itself) into an improved dense object detection paradigm, i.e., each voxel first regresses object borders, then uses this information to guide instance-level aggregation, and finally predicts class labels. The fundamental strength of our method is that each voxel can perceive whole object instances and aggregate instance-level geometry (context) to make a reliable class prediction, instead of relying solely on limited, independent local voxels.
>
> **Reformulation.** We provide a detailed analysis of each formulation to clarify its differences:
>
> - **[Formulation 1] Segmentation (previous SSC works): Each voxel only predicts its class.** This paradigm uniformly treats all voxels of the same class as a **single class group**, and is **entirely agnostic to instance-level geometry**, such as length, scale, and height. Hence, it risks incorrectly estimating object instance scales, such as extremely long cars, as evidenced by the qualitative results in Figs. 6 and 7 of the Appendix.
>
> - **[Formulation 2] Our vanilla dense object detection: Each voxel regresses the borders of object instances and predicts the class simultaneously [#1].** Using the regressed object instances, this paradigm can represent the scene with **multiple instance groups** by capturing instance-level geometry. This differs from the segmentation formulation, which treats all voxels of a class as a **single class group** and is agnostic to instance geometry. Although the classification branch is optimized in a segmentation-like manner [#1], this formulation is fundamentally different due to the unique instance-level regression.
>
> - **[Formulation 3] Our improved dense object detection: Each voxel first regresses the borders of object instances, then aggregates instance-level context, and finally predicts the class.** Building on vanilla dense detection, we introduce a technical improvement, i.e., using the regressed offsets to enhance class prediction with instance-level aggregation, making the formulation more principled and practical.
>
> We further present a detailed ablation study comparing these formulations. **To verify Formulation 1 (segmentation),** we deploy a 3D UNet following the common practice of existing SSC works on the 3D feature volume, then apply a classification head and optimize it with the standard SSC classification loss. **To verify Formulation 2 (our vanilla dense detection),** we follow [#1] by introducing an additional regression head to predict offsets, optimized with an L1 loss, consistent with our main paper. **To evaluate Formulation 3 (our improved dense detection),** we implement the instance-level aggregation in the classification branch to enhance instance-level semantics. We also report the **Full Model (VoxDet),** which further adds a Spatially-decoupled Voxel Encoder to improve task-specific representation learning.
>
> We have the following observations and conclusions. **(1)** Compared to the segmentation (Formulation 1), introducing regression (Formulation 2) yields a substantial 2.71-point increase in IoU, demonstrating the fundamental strength of the detection-based paradigm in geometry, consistent with our motivation and prior work [#1]. **(2)** Adding instance-level aggregation (Formulation 3) further boosts mIoU by 1.57 points, showing that our design meaningfully enhances the conventional detection framework. **(3)** The full VoxDet achieves the highest performance by improving task-specific representation learning for both classification and regression. **This evidence fully justifies that our detection-based formulation yields optimal SSC performance.**
>
> |           | Formulation 1            | Formulation 2              | Formulation 3              | Full Version        |
> |:---------:|:------------------------:|:--------------------------:|:--------------------------:|:-------------------:|
> | IoU   | 42.71                    | 45.42 (+2.71)              | 46.79 (+4.08)              | 47.36 (+4.65)       |
> | mIoU  | 16.28                    | 16.42 (+0.14)              | 17.85 (+1.57)              | 18.73 (+2.45)       |
>
> [#1] FCOS: Fully Convolutional One-Stage Object Detection (ICCV 2019)
>
> ## [W2] Clarify technical details
>
> Sorry for the confusion. We show the core code for this part below (`TPVDecoupleModule`). For classification and regression, we use **unshared** `TPVDecoupleModule` (one per feature level) to extract task-specific features, which are then sent to two unshared task-specific FPNs, respectively. This module is parameter-efficient, as justified in Tab. 5 of the main paper.
>
> - **Details about $Conv_{(T)}^{(P)}$.** We use a single convolution layer for each $Conv_{(T)}^{(P)}$. For different feature levels and tasks, these convolutions do not share weights, encouraging fully decoupled representations.
>
> - **Effective Decoupling.** In the below source code, the main decoupling effect is achieved by unshared `DeformConv2dTPV` instead of just simple convolution, i.e., **the classification and regression branches learn different spatial offsets** (`offset_xy/yz/zx`), guided independently by their respective task losses. The effectiveness of this spatial decoupling has been well-justified in [#2, #3]. Moreover, **in Fig. 6 of the main paper, our qualitative results show that the two branches learn largely different features, justifying our decoupling design**. In Tab. 4 of the main paper, we also show that replacing DeformConv with a standard convolution causes noticeable performance drops, highlighting the **essential role of our spatial decoupling**.
>
> - **FPN for classification and regression share the network architecture but do not share weights.** We have conducted a detailed ablation study in Tab. 4 of the main paper, finding that using unshared FPN weights for the two tasks benefits task-specific representations, consistent with the observations in [#2, #3].
>
> [#2] Revisiting the Sibling Head in Object Detector (CVPR 2020)
> [#3] Disentangle Your Dense Object Detector (ACM MM 2021)
> ```
> class TPVDecoupleModule(nn.Module):
>     def __init__(self, in_channels, mid_channels=None, kernel_size=1, padding=1,
>                  gn_groups=32, use_bias=False, adaptive_pooling=True):
>         super().__init__()
>         mid_channels = mid_channels or in_channels
>         self.adaptive_pooling = adaptive_pooling
>
>         if adaptive_pooling:
>             self.weights_conv = nn.Conv3d(in_channels, 3, 1, bias=use_bias)
>             self.pool_xy = DenseProjection('xy')
>             self.pool_yz = DenseProjection('yz')
>             self.pool_zx = DenseProjection('zx')
>
>         self.conv_xy = nn.Conv2d(in_channels, mid_channels, kernel_size, padding=padding)
>         self.conv_yz = nn.Conv2d(in_channels, mid_channels, kernel_size, padding=padding)
>         self.conv_zx = nn.Conv2d(in_channels, mid_channels, kernel_size, padding=padding)
>
>         offset_ch = 2 * 3 * 3
>         self.offset_xy = nn.Conv2d(mid_channels, offset_ch, 3, padding=1)
>         self.offset_yz = nn.Conv2d(mid_channels, offset_ch, 3, padding=1)
>         self.offset_zx = nn.Conv2d(mid_channels, offset_ch, 3, padding=1)
>
>         self.def_xy = DeformConv2dTPV(mid_channels, mid_channels, 3, padding=1, bias=False)
>         self.def_yz = DeformConv2dTPV(mid_channels, mid_channels, 3, padding=1, bias=False)
>         self.def_zx = DeformConv2dTPV(mid_channels, mid_channels, 3, padding=1, bias=False)
>
>         self.gn_xy = nn.GroupNorm(gn_groups, mid_channels)
>         self.gn_yz = nn.GroupNorm(gn_groups, mid_channels)
>         self.gn_zx = nn.GroupNorm(gn_groups, mid_channels)
>         self.act   = nn.ReLU(inplace=True)
>
>         self.fuse_conv = nn.Conv3d(mid_channels, in_channels, 1, bias=False)
>         self.fuse_gn   = nn.GroupNorm(gn_groups, in_channels)
>         self.fuse_act  = nn.ReLU(inplace=True)
>
>     def forward(self, x):
>         if self.adaptive_pooling:
>             w  = self.weights_conv(x)
>             xy = self.pool_xy(x, w)
>             yz = self.pool_yz(x, w)
>             zx = self.pool_zx(x, w)
>         else:
>             xy = x.mean(4); yz = x.mean(2); zx = x.mean(3)
>
>         # TPV spatial decoupling
>         def proc(f, conv, off, dconv, gn):
>             f = self.act(conv(f)) # Conv_{(T)}^{(P)}
>             o = off(f)
>             return self.act(gn(dconv(f, o))) # DefConv_{(T)}^{(P)}
>
>         dxy = proc(xy, self.conv_xy, self.offset_xy, self.def_xy, self.gn_xy)
>         dyz = proc(yz, self.conv_yz, self.offset_yz, self.def_yz, self.gn_yz)
>         dzx = proc(zx, self.conv_zx, self.offset_zx, self.def_zx, self.gn_zx)
>
>         B, C, X, Y, Z = x.shape
>         dxy = dxy.unsqueeze(-1).expand(-1, -1, -1, -1, Z)
>         dyz = dyz.unsqueeze(2 ).expand(-1, -1, X, -1, -1)
>         dzx = dzx.unsqueeze(3 ).expand(-1, -1, -1, Y, -1)
>
>         out = dxy + dyz + dzx
>         out = self.fuse_act(self.fuse_gn(self.fuse_conv(out)))
>
>         return out
> ```
>
> ## Response to tiny weakness
>
> Thank you for your careful review and constructive suggestions.
> 1. We will include the following related works [#4, #5], which investigate object instances in 2D space for SSC and further support the paradigm used in previous studies.
> 2. We will revise accordingly as suggested.
> 3. We underline “Vox” and “Det” in our claim to highlight the name of our method, VoxDet.
> 4. We will adjust the font size according to the final version.
>
> [#4] Disentangling Instance and Scene Contexts for 3D Semantic Scene Completion (ICCV 25)
> [#5] VLScene: Vision-Language Guidance Distillation for Camera-Based 3D Semantic Scene Completion (AAAI 25)

---

> > ### Comment · Area_Chair_K2nj · 2025-08-01
> >
> > Dear Reviewer ai3H,
> >
> > I've reviewed the authors' response, and they have provided detailed explanations addressing your two main concerns. Could you please review their response? Thank you.
> >
> > Best,
> >
> > AC

---

### Comment · Area_Chair_K2nj · 2025-08-09

Dear Reviewers,

According to this year's policy, the reviewer should discuss with the author or comment on the rebuttal before making the "mandatory acknowledgement." Please avoid marking the "mandatory acknowledgement" directly. Thank you.

Best,

AC

---

### Decision · Program_Chairs · 2025-09-17

**Decision:**

Accept (spotlight)

**Comment:**

This paper formulates semantic scene completion (SSC) as a dense object detection task, incorporating offset regression and semantic prediction to address 3D semantic scene completion.  The framework conducts two separate spatial decoupling modules, where one branch predicts voxel semantic labels while the other aims to capture instance-level information of each voxel. The learned instance-level voxels are integrated into the semantic-level voxels for scene completion. The work is based on the assumption that individual objects are separable in 3D so there is a change in semantic label at the object boundaries.  Experimental comparisons demonstrate that the proposed method achieves superior performance compared to other recent SSC methods.

The main strengths of this work are as follows: (1)  The idea is simple yet effective. The main insight of this work is the reformulation of SSC. The authors utilize two modules (e.g., spatially-decoupled voxel encoder and task-decoupled dense predictor) to achieve effective semantic scene completion, leading to improved SSC performance. (2) The experiments are quite strong. The work achieves state-of-the-art results on two semantic scene completion benchmarks.  (3) The paper is well-written, with proper content for high-level motivations, low-level technical details, and clear figures.

The main weakness of the paper is as follows:  (1) Novelty issue. The proposed method appears to be similar to previous segmentation-based methods. (2) The work is based on the assumption that there is a semantic label change on the object boundaries. This assumption does not always hold. (3) Missing some technical details. (4) Missing some details about the training process and the dataset.

This work provides a new perspective on the semantic scene completion (SSC) and demonstrates that using the object detection framework can outperform state-of-the-art SSC methods. All reviewers agree that this work is interesting and the experiments demonstrate that the method achieves superior performance compared to other recent SSC methods.

At the first round, the reviewers provide 5,5,4,3 scores. After the rebuttal, the rates become 6555 due to the main concerns about the assumption issues, and the detailed implementations are justified. I agree with the reviewers' comments and discussions. This work regards the SSC as a dense prediction or detection task. Inspired by previous dense prediction frameworks, the new SSC framework could achieve better performance than SOTA methods.  However, the final version should involve the discussions in the rebuttal to justify the assumption issues and the omission of some implementation details.